# OPTIMAL MULTIPLE TRANSPORT WITH APPLICATIONS TO VISUAL MATCHING, MODEL FUSION AND BEYOND

## ABSTRACT

Optimal transport (OT) has wide applications including machine learning. It concerns finding the optimal mapping for Monge OT (or coupling for Kantorovich OT) between two probability measures. This paper generalizes the classic pairwise OT to the so-called Optimal Multiple Transportation (OMT) accepting more than two probability measures as input. We formulate the problem as minimizing the transportation costs between each pair of distributions and meanwhile requiring cycle-consistency of transportation among probability measures. In particular, we present both the Monge and Kantorovich formulations of OMT and obtain the approximate solution with added entropic and cycle-consistency regularization, for which an iterative Sinkhorn-based algorithm (ROMT-Sinkhorn) is proposed. We empirically show the superiority of our approach on two popular tasks: visual multi-point matching (MPM) and multi-model fusion (MMF). In MPM, our OMT solver directly utilizes the cosine distance between learned features of points obtained from off-the-shelf graph matching neural networks as the pairwise cost. We leverage the ROMT-Sinkhorn algorithm to learn multiple matchings. For MMF, we focus on the problem of fusing three models and employ ROMT-Sinkhorn instead of the Sinkhorn algorithm to learn the alignment between layers. Both tasks achieve competitive results with ROMT-Sinkhorn. Furthermore, we showcase the potential of our approach in addressing the travel salesman problem (TSP) by searching for the optimal path on the probability matrix instead of the distance matrix. Source code will be made publicly available.

## 1 INTRODUCTION

Optimal transport (OT) (Peyre & Cuturi, 2019) is a mathematical tool with wide applications including vision and learning. In general, it aims to learn the optimal transportation between the source and target probability measures. While mainstream OT research primarily focuses on transportation between two distributions, less attention has been given to studying the general setting when there are multiple distributions for transportation. In fact, many real-world problems ranging from point matching (Wang et al., 2023) to model fusion (Liu et al., 2022a) often involve more than two distributions or sets and there calls for a unified approach, beyond naively running pairwise transportation in isolation.

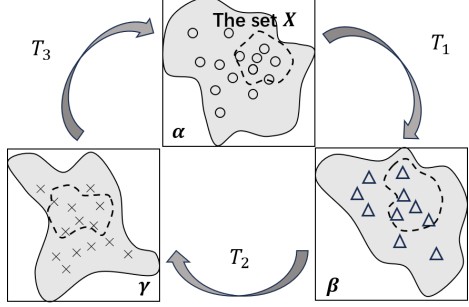

Figure 1: An illustration of Optimal Multiple Transport (OMT). Assuming three (or more) probability measures $\alpha, \beta, \gamma$, OMT satisfies $T_{1\#}\alpha = \beta, T_{2\#}\beta = \gamma, T_{3\#}\gamma = \alpha$, and the cycle-consistency constraints $X = T_3(T_2(T_1(X)))$ given a set $X$ sampled from $\alpha$.

With the above setting, in this paper, we study the generalized form of OT for more than two probability measures and term it Optimal Multiple Transport (OMT). As illustrated in Fig. 1, the case considers three probability measures and it seeks to minimize the cost of three transportation while ensuring cycle-consistency among measures: specifically given a set $X$ sampled from the probability measure $\alpha$, the transportation mappings $T_1, T_2, T_3$ satisfy the condition $X = T_3(T_2(T_1(X)))$. In this setting, we propose Monge and Kantorovich formulations for OMT, considering the challenges posed by

cycle-consistency constraints in solving the problems. In this paper, we introduce the entropic regularizer and transform the hard cycle-consistency constraint into a regularization term. This transformation leads to a regularized version of OMT, as approximately and efficiently solved via our proposed tailored iterative Sinkhorn algorithm called ROMT-Sinkhorn algorithm.

Furthermore, based on OMT, we discover a side product: a new and easily comprehensible formulation for the Traveling Salesman Problem (TSP), which we refer to as TSP-OMT. In this formulation, we construct closed-loop circuits for TSP using the cycle-consistency constraint. To compute an approximate solution for TSP-OMT, the regularized approach is to get the approximated solution but fail to guarantee closed-loop circuits with the probability matrix. However, this inspired us to explore searching for paths in the probability space rather than the distance space. To compare the solutions, we employ a greedy algorithm to find TSP paths in the Euclidean space, Sinkhorn probability space, and the regularized TSP-OMT probability space and find our method outperforms, which shows the potential to construct a probability space from the distance.

In our experiments, we provide empirical case studies concerning two popular matching tasks: visual multi-point matching and neural multi-model fusion. For the former task, we compute a certain node-to-point distance e.g. cosine distance using the learned node feature from graph matching networks (e.g. (Wang et al., 2021))[1] and adopt ROMT-Sinkhorn for inference these neural matching model. For the latter task, we also adopt the ROMT-Sinkhorn instead of the pairwise Sinkhorn algorithm to calculate the layer alignment for the fusion of three models. Both two task experiments achieve competitive results demonstrating the effectiveness and superiority of our proposed approach. **The highlights of this paper are as follows:**

1) We generalize OT to the multi-party case called Optimal Multiple Transport (OMT), which solves multiple transportation mappings between probability measures while try to ensure cycle-consistency among these mappings. Both its Monge and Kantorivich formulations are developed.

2) We model the OMT problem by adopting entropic and cycle-consistency regularization, and propose an iterative Sinkhorn algorithm named ROMT-Sinkhorn, to obtain an approximate solution.

3) As a side product, we introduce a new formulation for the Traveling Salesman Problem (TSP) called TSP-OMT, in which we incorporate cycle-consistency to capture the closed-loop constraint of TSP. We use the regularized TSP-OMT formulation to compute the probability matrix of TSP, which allows us to search for the optimal path in the probability space instead of the distance space.

4) We apply the ROMT-Sinkhorn algorithm to two domains: multi-point matching and multi-model fusion. The competitive experimental results show the superiority of our methods.

## 2 RELATED WORKS AND PRELIMINARIES

### 2.1 OPTIMAL TRANSPORTATION

Given two probability measures $\alpha$ and $\beta$ supported on $\mathcal{X}$ and $\mathcal{Y}$, the Monge formulation of Optimal Transportation (Monge, 1781) aims to find a mapping $T : \mathcal{X} \to \mathcal{Y}$ that minimizes an overall cost:

$$\min_T \{ \int_{\mathcal{X}} c(x, T(x))d\alpha(x) : T_{\#}\alpha = \beta \} \tag{1}$$

where $c(\cdot, \cdot)$ is the cost function and the push-forward measure $\beta = T_{\#}\alpha$ means the satisfaction $\beta(\mathcal{S}) = \alpha(x \in \mathcal{X} : T(x) \in \mathcal{S})$, for an arbitrary set $\mathcal{S} \subset \mathcal{Y}$. The Monge problem is exactly not easy to calculate and a popular improvement is the Kantorovich relaxation (Kantorovich, 1942) which seeks the coupling $\mathbf{P}$ instead. Specifically, for the discrete case, we assume $\alpha = \sum_{i=1}^n \mathbf{a}_i \delta_{x_i}$ and $\beta = \sum_{j=1}^m \mathbf{b}_j \delta_{y_j}$ where $(\{x_i\}, \{y_j\})$ are the locations from $(\mathcal{X}, \mathcal{Y})$, and $(\mathbf{a}, \mathbf{b})$ are probability vectors. Then the Kantorovich problem aims to find the coupling $\mathbf{P}$, which is specified as

$$\min_{\mathbf{P} \in U(\mathbf{a}, \mathbf{b})} < \mathbf{C}, \mathbf{P} > = \sum_{ij} \mathbf{C}_{ij} \mathbf{P}_{ij}, \tag{2}$$

where $U(\mathbf{a}, \mathbf{b}) = \{\mathbf{P} \in R_{nm}^+ | \mathbf{P}\mathbf{1}_m = \mathbf{a}, \mathbf{P}^\top \mathbf{1}_n = \mathbf{b}\}$ and $\mathbf{C}$ is the cost matrix defined by the divergence between $\{x_i\}_{i=1}^n$ and $\{y_j\}_{j=1}^m$. This minimization can link to the linear program (Bertsimas &

---

[1]Note that the visual graph matching networks in fact embed the structure information into the node features hence their output is node-wise features suitable for our OT setting.

Tsitsiklis, 1997) but the calculation speed is really slow for high dimensions. Entropic regularization (Cuturi, 2013) is one of the simple but efficient methods for solving OT problems:

$$\min_{\mathbf{P} \in U(\mathbf{a}, \mathbf{b})} < \mathbf{C}, \mathbf{P} > -\epsilon H(\mathbf{P}), \text{ where } H(\mathbf{P}) = - < \mathbf{P}, \log \mathbf{P} - \mathbf{1}_{m \times n} > . \tag{3}$$

Here $\epsilon > 0$ is the regularization coefficient. This entropic OT can be solved by Sinkhorn iterations by vector-matrix multiplication (Cuturi, 2013).

**Multi-marginal Optimal Transport.** Instead of coupling two histograms $(\mathbf{a}, \mathbf{b})$ in Kantorovich problem, the multi-marginal optimal Transportation couples $K$ histograms $(\mathbf{a}^k)_{k=1}^K$ by solving the following multi-marginal transport (Abraham et al., 2017):

$$\min_{\mathbf{P} \in U((\mathbf{a}^k)_k)} < \mathbf{C}, \mathbf{P} > = \sum_k \sum_{i_k=1}^{n_k} \mathbf{C}_{i_1, i_2, \dots, i_K} \mathbf{P}_{i_1, i_2, \dots, i_K} \tag{4}$$

where $\mathbf{C}_{i_1, i_2, \dots, i_K}$ is $n_1 \times \cdots \times n_K$ cost tensor and the valid coupling set $U((\mathbf{a}^k)_k)$ is defined as

$$U((\mathbf{a}^k)_k) = \{\mathbf{P} \in \mathbb{R}^+_{n_1 \times n_2 \dots n_K} | \forall k, \forall i_k, \sum_{l \neq k} \sum_{i_l=1}^{n_l} \mathbf{P}_{i_1, \dots, i_K} = \mathbf{a}^k_{i_k}\}. \tag{5}$$

The multi-marginal OT and our OMT both deal with multiple distributions. However, the multi-marginal OT primarily emphasizes learning the joint coupling among more than two distributions, whereas our focus is on learning the coupling between each pair of distributions and maintaining cycle-consistency constraints among these couplings.

## 2.2 CYCLE CONSISTENCY FOR MATCHING

The idea of cycle consistency has been broadly considered in learning and vision. Examples include the application in multiple graph matching (Wang et al., 2021; Bernard et al., 2019; Tourani et al., 2023), image matching (Sun et al., 2023; Bernard et al., 2019), and shape matching (Bhatia et al., 2023; Bernard et al., 2019) etc. These instances of multiple matching with cycle-consistency in various domains motivate us to investigate whether multiple transportation can be performed with cycle-consistent constraints in the Optimal Transport problem. Thus, in this paper, we elaborate on the concept of cycle-consistency in OT and introduce the definition of OMT in Sec. 3.1.

## 2.3 VISUAL MATCHING AND MODEL FUSION

**Visual Point Matching** (PM) (Sarlin et al., 2020; Sun et al., 2021) is a significant research area in computer vision that aims to find optimal point correspondences between images, which has various applications, such as 3D structure estimation and camera pose estimation. Graph matching (GM) (Caetano et al., 2009) builds upon PM and treats the point sets as graphs, aiming to find the optimal node correspondences between graph-structured data. GM can be typically formulated as LAP (Crama & Spieksma, 1992), which is known to be NP-hard and requires expensive and complex solvers. Recent works (Wang et al., 2019; Yu et al., 2019) have focused on learning node features using supervised or unsupervised loss functions. In this paper, our main focus is on multi-point matching (Swoboda et al., 2019), where we utilize the trained models (Wang et al., 2019) to extract point features and perform inference on testing data, which emphasizes the cycle-consistency among multiple images, enabling more robust and accurate matching results.

**Model fusion** (for neural networks) as a post-processing step after model training, has gained increasing attention, with methods ranging from optimal transport (Singh & Jaggi, 2020; Imfeld et al., 2023) to graph matching (Liu et al., 2022b), which permute the neural weights of each model for alignment (i.e. fusion). Besides, it is also proposed for merging different neural network architectures (Wang et al., 2020a), heterogeneous neural networks (Nguyen et al., 2023) and models of different tasks (Stoica et al., 2023). (Wortsman et al., 2022) show that averaging the weights of models fine-tuned with different hyperparameter configurations often improves accuracy and robustness for large pretrained models and (Matena & Raffel) propose a "Fisher merging" method that provides a performance boost in settings with simple parameter averaging. In this paper, we focus on improving alignments for multiple models and show how to leverage the cycle-consistency in our OMT framework compared with the pairwise fusion baseline (Singh & Jaggi, 2020).

## 3 OPTIMAL MULTIPLE TRANSPORTATION AND OUR SOLVERS

In this section, we begin by presenting the Monge and Kantorovich formulations for Optimal Multiple Transport (OMT) in Sec. 3.1. Then, we introduce regularized terms that are incorporated into OMT,

leading to the development of the iterative Sinkhorn algorithm (called ROMT-Sinkhorn Algorithm) in Sec. 3.2. Lastly, in Sec. 3.3, we leverage the principles of OMT to devise a novel formulation of the Traveling Salesman Problem (TSP) that highlights the theoretical potential of OMT.

## 3.1 THE MONGE AND KANTOROVICH FORMULATION OF OMT

**OMT's Monge Formulation.** We first assume $K$ probability measures $(\alpha_k)_{k=1}^K$ supported on the space $(\mathcal{X}_k)_{k=1}^K$. Note for simplicity, we define that $\mathcal{X}_{K+1} = \mathcal{X}_1$ and $\alpha_{K+1} = \alpha_1$. Then OMT aims to find mappings $(T_k)_{k=1}^K$ where $T_k : \mathcal{X}_k \to \mathcal{X}_{k+1}$ by optimizing the objective function that

$$\min_{(T_k)_k \in \mathcal{C}((\alpha_k)_k)} \sum_i \int_{\mathcal{X}_k} c_k(x, T_k(x)) d\alpha_k(x), \tag{6}$$

where $c_k(\cdot, \cdot)$ is the cost function for the space $(\mathcal{X}_k, \mathcal{X}_{k+1})$. The constraint $\mathcal{C}((\alpha_k)_k)$ is specified:

$$\mathcal{C}((\alpha_k)_k) = \{(T_k)_{k=1}^K | (T_k)_\# \alpha_k = \alpha_{k+1}, \forall k; \ T_k \circ T_{K-1} \circ \cdots \circ T_2 \circ T_1(X) = X, \forall X \subset \mathcal{X}_1\}, \tag{7}$$

where $(T_k)_\# \alpha_k = \alpha_{k+1}$ is the push-forward operation from measure $\alpha_k$ to $\alpha_{k+1}$ satisfying $\alpha_{k+1}(B \in \mathcal{X}_{k+1}) = \alpha_k(x \in \mathcal{X}_k | T_k(x) \in B)$ for any measurable set $B$. And $\forall X \subset \mathcal{X}_1$, the equality $T_k \circ T_{K-1} \circ \ldots T_2 \circ T_1(X) = X$ is the cycle-consistency constraint that enforces the final transport results aligning to the original one beginning at points in $\mathcal{X}_1$. Naturally, we can get the measure $\alpha_1(X) = \alpha_1(T_k \circ \cdots \circ T_1(X))$. Note the cycle-consistency starts from $\alpha_1$ and one can also formulate the OMT's Monge problem starting from $\alpha_2, \alpha_3, \ldots, \alpha_K$. For the calculation, the OMT's Monge formulation encounters difficulties like those of traditional Monge OT and the solution may even not exist in discrete cases. Hence for OMT, Kantorovich relaxation is introduced for tractability.

**OMT's Kantorovich Formulation.** Assume that the (probability) measures $\alpha_k = \sum_{i=1}^{n_k} \mathbf{a}_i^k \delta_{x_i^k}$ where $x_i^k$ is the location in $\mathcal{X}_k$ space, and $\mathbf{a}^k$ is the histogram for $\alpha_k$. Then the Kantorovich OMT aims to seek $K$ coupling matrices $(\mathbf{P}_k)_{k=1}^K$ where $\mathbf{P}_k$ is the coupling between $\alpha_k$ and $\alpha_{k+1}$:

$$\min_{(\mathbf{P}_k)_k \in \mathcal{C}'((\mathbf{a}^k)_k)} \sum_{k=1}^K < \mathbf{C}_k, \mathbf{P}_k >, \tag{8}$$

where $\mathbf{C}_k$ is the cost matrix between locations $(x_i^k)_i$ and $(x_i^{k+1})_i$, and the coupling set $\mathcal{C}((\mathbf{a}^k)_{k=1}^K)$ is

$$\mathcal{C}'((\mathbf{a}^k)_{k=1}^K) = \left\{ (\mathbf{P}_k)_{k=1}^K | \forall k, \mathbf{P}_k \in U(\mathbf{a}^k, \mathbf{a}^{k+1}), \prod_{k=1}^K \tilde{\mathbf{P}}_k = \mathbf{I} \right\}. \tag{9}$$

Here $\mathbf{I}$ represents the identity matrix, and $\tilde{\mathbf{P}}_k = \mathbf{P}_k \oslash \mathbf{a}^k$ is obtained through element-wise division $\oslash$, which signifies row normalization of the matrix $\mathbf{P}_k$ (i.e., $\tilde{\mathbf{P}}_k = \text{Diag}(1/\mathbf{a}^k)\mathbf{P}_k$). Note the constraints $\prod_{k=1}^K \tilde{\mathbf{P}}_k = \mathbf{I}$ aim to ensure cycle-consistency. Specifically, for any measure $\alpha'$ supported on $\mathcal{X}_1$ with probability vector $\mathbf{a}'$, we consider $(\mathbf{a}'\tilde{\mathbf{P}}_1)$ as the transition from $\mathcal{X}_1$ to $\mathcal{X}_2$ with the matrix $\tilde{\mathbf{P}}_1$ satisfying $\sum_i \mathbf{a}_i' = \sum_i (\mathbf{a}'\tilde{\mathbf{P}}_1)_i$. Similarly, We can transport the $\mathbf{a}'$ to $\mathcal{X}_2, \ldots, \mathcal{X}_K$ space with $\tilde{\mathbf{P}}_2, \ldots, \tilde{\mathbf{P}}_K$. Then $\mathbf{a}' \prod_{k=1}^K \tilde{\mathbf{P}}_k = \mathbf{a}'$ represents the probability $\mathbf{a}'$ being transported back to its initial state. Consequently, we can naturally derive the constraint $\prod_{k=1}^K \tilde{\mathbf{P}}_k = \mathbf{I}$ as the cycle-consistency constraint. In essence, we can understand $\tilde{\mathbf{P}}_k$ as a probability transition matrix in a Markov chain, where the sum of probability transition elements in each row is equal to 1, and $\prod_{k=1}^K \tilde{\mathbf{P}}_k$ represents the $K$-step transition probability matrix, which returns to the identity matrix as the cycle-consistency.

For Eq. 8, note the Kantorovich form of OMT is no longer a linear programming due to the constraints of cycle-consistency. For efficiency, we propose the regularized OMT, which allows to derive an iterative Sinkhorn algorithm for obtaining approximate solutions.

## 3.2 REGULARIZED OMT AND THE SOLVING ALGORITHM

By applying entropic regularization to relax Eq. 8 (Cuturi, 2013), we transform the cycle-consistency into a regularizer which is Specifically formulated as:

$$\min_{(\mathbf{P}_k)_k : \mathbf{P}_k \in U(\mathbf{a}^k, \mathbf{a}^{k+1})} \sum_{k=1}^K < \mathbf{C}_k, \mathbf{P}_k > -\epsilon \sum_k H(\mathbf{P}_k) - \delta || \prod_{k=1}^K \tilde{\mathbf{P}}_k - \mathbf{I}||_F^2 \tag{10}$$

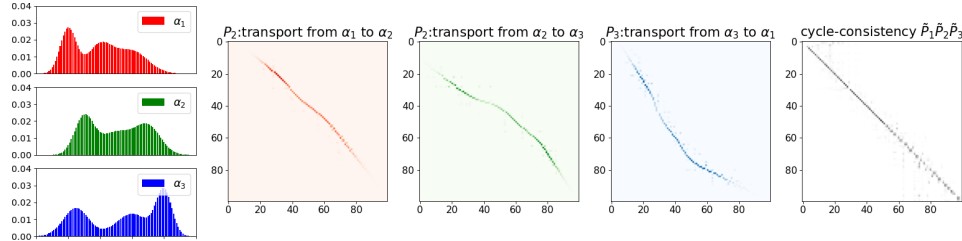

Figure 2: Illustration of transport solutions $\mathbf{P}_1$, $\mathbf{P}_2$, and $\mathbf{P}_3$, along with the cycle-consistency matrix $\tilde{P}_1\tilde{P}_2\tilde{P}_3$, based on three given histograms. The histograms on the left correspond to $\alpha_1$, $\alpha_2$, and $\alpha_3$, while the middle histograms represent the couplings from $\alpha_1$ to $\alpha_2$, $\alpha_2$ to $\alpha_3$, and $\alpha_3$ to $\alpha_1$. Finally, the rightmost matrix denotes $\tilde{P}_1\tilde{P}_2\tilde{P}_3$, which exhibits a close similarity to the identity matrix.

where $\epsilon, \delta > 0$ are the coefficients for entropic and cycle-consistent regularization respectively. By introducing Lagrangian dual variables $(\mathbf{f}_k, \mathbf{g}_k)$ for each marginal constraint in $U(\mathbf{a}^k, \mathbf{a}^{k+1})$ with $k = 1, 2, \ldots, K$, we can get equation with respect to $\mathbf{P}_k$ as follows:

$$\frac{\partial \mathcal{L}}{\partial \mathbf{P}_k} = \mathbf{C}_k + \epsilon \log \mathbf{P}_k - \mathbf{f}_k \mathbf{1}^\top - \mathbf{1}\mathbf{g}_k^\top - \delta M_k = 0 \tag{11}$$

where $\mathcal{L}$ is the Lagrangian function and $M_k$ is specified as

$$M_k = 2\text{Diag}\left(\frac{1}{\mathbf{a}^k}\right)\left(\prod_{t_1=1}^{k-1} \tilde{\mathbf{P}}_{t_1}\right)^\top \left(\prod_{t_2=1}^{K} \tilde{\mathbf{P}}_{t_2} - \mathbf{I}\right)\left(\prod_{t_3=k+1}^{K} \tilde{\mathbf{P}}_{t_3}\right)^\top. \tag{12}$$

According to Eq. 11, the solution has the form

$$\mathbf{P}_k = \text{Diag}(e^{\mathbf{f}_k/\epsilon})e^{(-\mathbf{C}_k+\delta M_k)/\epsilon}\text{Diag}(e^{\mathbf{g}_k/\epsilon}) \tag{13}$$

The details are given in Appendix A. When $\delta = 0$, the solution $\mathbf{P}_k$ degenerates into the vanilla entropic OT and when $\delta > 0$, $(\mathbf{P}_k)_k$ tend to satisfy cycle-consistency. Note the matrices $(M_k)_k$ are derived from the probability matrices $\mathbf{P}_k$. If the matrices $(\mathbf{P}_k)_k$ are known, we can update $(M_k)_k$ using Eq. 12. Similarly, if $(M_k)_k$ are known, $(\mathbf{P}_k)_k$ can be updated via the Sinkhorn algorithm. By initializing the matrices $(M_k^{(0)})_k$ as zero matrices, we can obtain $(\mathbf{P}_k^{(0)})_k$ as the pairwise Sinkhorn solutions. Then, by iteratively updating $(M_k^{(l)})_k$ and $(\mathbf{P}_k^{(l)})_k$ till convergence by Eq. 12 and by $\mathbf{P}_k^{l+1} = \text{Sinkhorn}(\mathbf{C}_k - \delta M_k^{(l)}, \mathbf{a}^k, \mathbf{a}^{k+1})$, we can obtain the solution of Regularized OMT in Eq. 10.

We present our algorithm in Algorithm 1. As shown in Fig. 3, 6 points are sampled from three 2D-Gaussian distributions and Euclidean distances are used as costs for computing couplings. Compared to the pair-wise Sinkhorn algorithm, our ROMT-Sinkhorn achieves cycle-consistency results. Fig. 2 illustrates the transportation results among more complex distributions. It is noteworthy that the left three histograms are sampled from Gaussian mixture distributions, and the couplings can be computed using the ROMT-Sinkhorn algorithm as shown in the middle three subfigures. As shown in the rightmost subfigure, the cycle-consistency $\prod_{k=1}^{K} \tilde{\mathbf{P}}_k = \mathbf{I}$ is almost satisfied.

**Transportation's Order Switching Problem.** For the order of sets, assume $K = 3$ and the couplings are $\mathbf{P}_1$, $\mathbf{P}_2$, and $\mathbf{P}_3$. Consistency requires $\mathbf{P}_1\mathbf{P}_2\mathbf{P}_3 = \mathbf{I}$. If we switch the second and third orders with matching matrices $\mathbf{P}_3^T$, $\mathbf{P}_2^T$, and $\mathbf{P}_1^T$, the consistency becomes $\mathbf{P}_3^T\mathbf{P}_2^T\mathbf{P}_1^T = \mathbf{P}_1\mathbf{P}_2\mathbf{P}_3 = \mathbf{I}$. Thus, for $K = 3$, switching the order does not affect the problem formulation. As for $K > 3$, switching the order does indeed impact the formulation of the problem. However, note our directed cyclical structure is essentially a subgraph of pairwise structure. When the latter is satisfied, the former is also satisfied, allowing our method to still improve the matching performance.

### 3.3 Side Product: A new OMT-based TSP Formulation

From the multiple transportation view, we consider an interesting assumption: What if all the transportation is in the same space? This means we can set that $\alpha = \alpha_1 = \alpha_2 = \cdots = \alpha_K$ and then all probability measures share the same locations and histograms, which leads to the same cost matrix (i.e. $\mathbf{C} = \mathbf{C}_1 = \mathbf{C}_2 = \cdots = \mathbf{C}_K$) and coupling solutions (i.e. $\mathbf{P} = \mathbf{P}_1 = \mathbf{P}_2 = \cdots = \mathbf{P}_K$) for all transportation. Under this assumption, the cycle-consistency is transformed into $\mathbf{P}^K = \mathbf{I}$, which

**Algorithm 1** ROMT-Sinkhorn: Iterative Sinkhorn-based Algorithm for Regularized OMT

**Input:** Cost Matrices $(\mathbf{C}_k)_{k=1}^K$ and histograms $(\mathbf{a}^k)_{k=1}^K$, iteration number $L$

**Output:** the couplings $(\mathbf{P}_k^{(L-1)})_{k=1}^K$

   Initialize $M_k^{(0)} = \mathbf{0}_{n_k, n_{k+1}}$    for all $k$

   **for** $l = 0, 1, \ldots, L - 1$ **do**

      **for** $k = 1, 2, \ldots, K$ **do**

         $\mathbf{P}_k^{(l)} = \text{Sinkhorn}(\mathbf{C}_k - \delta M_k^{(l)}, \mathbf{a}^k, \mathbf{a}^{k+1})$

         $\tilde{\mathbf{P}}_k^{(l)} = \mathbf{P}_k^{(l)} \oslash \mathbf{a}^k$

      **end for**

      Calculate $(M_k^{(l+1)})_k$ by Eq. 12 with $(\tilde{\mathbf{P}}_k^{(l)})_k$

   **end for**

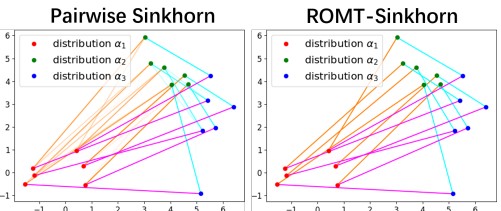

Figure 3: Multiple Matching on 2-D points. The left one is the result of pairly adopting the Sinkohrn algorithm; the right is the solutions of our ROMT-Sinkhorn Algorithm 1. We can find our matching forms a closed loop, whereas the pairwise Sinkhorn results do not.

implies that each point returns to its original location after transportation. This inspires us to draw a connection to the Traveling Salesman Problem (TSP).

In TSP, it assumes $K$ points with its distance matrix $\mathbf{C}$ and our aim is to get the solution $\mathbf{P} \in \{0, 1\}^{K \times K}$. Note we assume the transport for points themselves is not allowed (i.e. $(\mathbf{C})_{ii} \to \infty$ and thus $(\mathbf{P})_{ii} = 0$). For simplicity, we **break the probability measure assumption** (i.e. $\sum_i \mathbf{a}_i^k = 1$) and set $\mathbf{a}^k = \mathbf{1}$ for all $k$ to coincide with TSP. Besides, we apply the **cycle-consistency** view in OMT to capture the **closed-loop** constraint, which forms a new TSP formulation (called TSP-OMT) as:

$$\min_{\mathbf{P}} <\mathbf{C}, \mathbf{P}> \text{ s.t. } \mathbf{P}\mathbf{1}_K = \mathbf{1}_K, \ \mathbf{P}^\top \mathbf{1}_K = \mathbf{1}_K, \ <\mathbf{P}^k, \mathbf{I}> = 0(\forall k < K), \ \mathbf{P}^K = \mathbf{I}. \quad (14)$$

Here, $\mathbf{P}^k$ represents the k-th power of matrix $\mathbf{P}$ and the condition $<\mathbf{P}^k, \mathbf{I}> = 0$ for $k < 0$ is introduced to terminate the consistency process before the final step to ensure that $(\mathbf{P}^k)_{ii} = 0$, which guarantees that the probability of a traveling salesman starting from position $i$, taking $k$ steps $(k < K)$, and returning to position $i$ is zero. On the other hand, the condition $(\mathbf{P})^K = \mathbf{I}$ is imposed to enforce cycle-consistency, ensuring that $(\mathbf{P}^K)_{ii} = 1$, which guarantees that the salesman returns to their original position, completing the cycle.

Note that the optimization in Eq. 14 is no longer a Linear Program problem. Similar to the approach in Sec. 3.2, the entropic and closed-loop regularization is employed for the minimization as:

$$\min_{\mathbf{P}} <\mathbf{C}, \mathbf{P}> -\epsilon H(\mathbf{P}) + \sum_k \delta_k ||\mathbf{P}^k - \mathbf{I}||_F^2 \quad \text{s.t.} \quad \mathbf{P}\mathbf{1}_K = \mathbf{1}_K, \ \mathbf{P}^\top \mathbf{1}_K = \mathbf{1}_K, \quad (15)$$

where $(\delta_k)_k$ are the regularization coefficients. We set $\delta_k < 0$ for $k < K$ to make $(\mathbf{P}^k)_{ii}$ approach 0 for every $k < K$, and $\delta_K > 0$ to make $(\mathbf{P}^k)_{ii}$ approach 1. Then with the Lagrangian method for Eq. 15, we can get the solution form as

$$\mathbf{P} = \text{Diag}(e^{\mathbf{f}/\epsilon}) e^{-(\mathbf{C}+M)/\epsilon} \text{Diag}(e^{\mathbf{g}/\epsilon}), \quad \text{and} \quad M = \sum_{k=1}^K \sum_{t=0}^{k-1} 2\delta_k (\mathbf{P}^t)^\top (\mathbf{P}^k - \mathbf{I}) \mathbf{P}^{k-1-t})^\top, \quad (16)$$

where $\mathbf{f}$ and $\mathbf{g}$ are Lagrangian duals. Then we can obtain the approximate solution of TSP with the iterative Sinkhorn algorithm as proposed in Algorithm 2. The details are given in Appendix B. However, unfortunately, Algorithm 2 can not achieve the ideal closed-loop solution which may be due to the simple setting of $\delta_k$ and too many regularized terms of closed-loop constraints.

In fact, our probability-based results on the other hand enable the selection of the TSP path from a probabilistic perspective rather than relying solely on the traditional distance matrix. This shift transforms TSP into a sampling problem, where the calculated probability matrix can be utilized. For example, we can employ a greedy method, as described in Appendix B, to search for a closed-loop path based on the probability matrix computed using Algorithm 2. In Fig. 4, 25 points are randomly sampled as the locations and we compare the total cost based on greedy

**Algorithm 2** Probability Matrix Calculation for Regularized TSP-OMT.

**Input:** Cost Matrix $\mathbf{C}$ and iteration number $L$

**Output:** the coupling $\mathbf{P}^{(L)}$

   Initialize $M^{(0)} = \mathbf{0}_{K \times K}$

   **for** $l = 0, 1, \ldots, L$ **do**

      $\mathbf{P}^{(l)} = \text{Sinkhorn}(\mathbf{C} + M^{(l)}, \mathbf{1}_K, \mathbf{1}_K)$

      Calculate $M^{(l+1)}$ by Eq. 16 with $\mathbf{P} = \mathbf{P}^{(l)}$

   **end for**

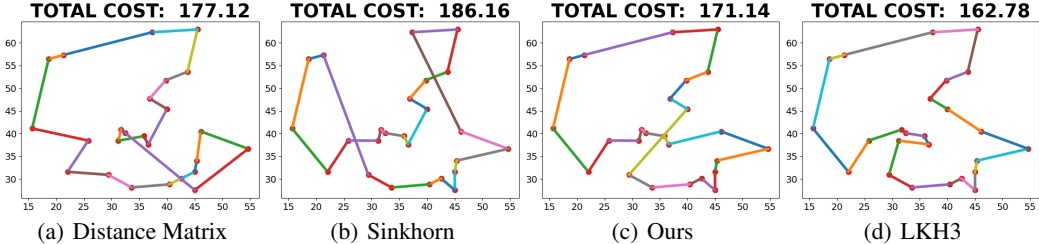

Figure 4: Comparison of TSP solutions. The left three adopt the greedy method on distance matrix, and probability matrix with Sinkhorn and our algorithm in Algorithm 2, respectively. The rightmost one is the strong TSP solver LKH3 (Helsgaun, 2017). Our method performs competitively.

search using the Euclidean distance matrix, Sinkhorn probability matrix, and our probability matrix calculated by Algorithm 2 in Fig. 4 and find that our approach performs competitively.

Though our current method is still far from competing with strong classic TSP solvers like LKH3 (Helsgaun, 2017), as shown in Fig. 4, it provides a new perspective for tackling the TSP, which involves converting the distance matrix into a probability matrix and searching for the optimal path based on the probabilities[2]. In the probability matrix, the edge selection is based on global considerations, which inherently provides an advantage over distance-based edge selection. Moreover, if an improved algorithm can be developed to obtain the closed-loop probability solution in the future, we would no longer need the sampling algorithm to determine the TSP path. This opens up possibilities for efficiently solving large-scale TSP problems using matrix scaling methods via GPU computing.

### 3.4 APPLICATION TO MULTI-POINT MATCHING AND MULTI-MODEL FUSION

In this subsection, we apply our method to the task of multi-set matching and multi-model fusion.

**OMT for Multi-point Matching.** Here, we apply the ROMT-Sinkhorn algorithm to the inference of the multi-point matching model. We assume the existence of multiple sets, each containing several point features extracted from images by the trained neural model. Our goal is to establish cycle-consistent matches among these sets. Specifically, given $K$ probability measures $(\alpha_k)_{k=1}^K$, where $\alpha_k = \sum_{i=1}^n \mathbf{a}_i^k \delta_{x_i^k}$, and $x_i^k$ represents the point feature, we can define the cost $\mathbf{C}_k$ between $\alpha_k$ and $\alpha_{k+1}$. Then the inference during the testing process can be formulated with Eq. 10. To solve the optimization, we utilize the ROMT-Sinkhorn Algorithm, as presented in Algorithm 1, to obtain predictions for the testing data. Fig. 5 illustrates the inference results using Pairwise Sinkhorn and ROMT-Sinkhorn algorithms, where a neural matching model (NMGM (Wang et al., 2021)) serves as the backbone. It can be observed that the coupling $\mathbf{P}_3$ generated by the pairwise Sinkhorn method contains a mismatch for two points at the rear of the vehicle. However, our ROMT-Sinkhorn Algorithm corrects this misalignment and produces accurate matching.

**OMT for Multi-model fusion.** Following (Singh & Jaggi, 2020) that applies the OT for model fusion task, we apply our ROMT-Sinkhorn algorithm instead of the previous pairwise Sinkhorn algorithm in (Singh & Jaggi, 2020) for multi-model fusion. Without loss of generality, here we consider fusing three models. Assume $\mathbf{W}_k^{(l,l-1)}$ is the weight matrix for model $k$ ($k = 1, 2, 3$) between layer $l$ and $l-1$, and $\widehat{\mathbf{W}}_k^{(l,l-1)}$ ($k = 2, 3$) is the modified weights with alignments $\tilde{\mathbf{P}}_1^{l-1}, \tilde{\mathbf{P}}_3^{l-1}$ before layer $l$:

$$\widehat{\mathbf{W}}_2^{(l,l-1)} = \mathbf{W}_2^{(l,l-1)}(\tilde{\mathbf{P}}_1^{(l-1)})^\top \quad \text{and} \quad \widehat{\mathbf{W}}_3^{(l,l-1)} = \mathbf{W}_3^{(l,l-1)}\tilde{\mathbf{P}}_3^{(l-1)}. \quad (17)$$

Then we can get the weight alignments for $\widetilde{\mathbf{W}}_2^{(l,l-1)}$ and $\widetilde{\mathbf{W}}_3^{(l,l-1)}$ to $\mathbf{W}_1^{(l,l-1)}$ by

$$\widetilde{\mathbf{W}}_2^{(l,l-1)} = \tilde{\mathbf{P}}_1^l \widehat{\mathbf{W}}_2^{(l,l-1)} \quad \text{and} \quad \widetilde{\mathbf{W}}_3^{(l,l-1)} = (\tilde{\mathbf{P}}_3^l)^\top \widehat{\mathbf{W}}_3^{(l,l-1)}, \quad (18)$$

where $\tilde{\mathbf{P}}_1^l$ is the alignment from model 1 to model 2 and $\tilde{\mathbf{P}}_3^l$ is the alignment from model 3 to model 1 for layer $l$ calculated by ROMT-Sinkhorn. Finally, we get the parameter matrix of the fused model:

$$\mathbf{W}_{\mathcal{F}}^{(l,l-1)} = \frac{1}{3}\Big(\mathbf{W}_1^{(l,l-1)} + \widetilde{\mathbf{W}}_2^{(l,l-1)} + \widetilde{\mathbf{W}}_3^{(l,l-1)}\Big). \quad (19)$$

Initializing $l = 2$ and updating $\widehat{\mathbf{W}}_k^{(l,l-1)}$ and $\widetilde{\mathbf{W}}_k^{(l,l-1)}$ (k=2,3) by varying $l$, we can get the fused model's parameter matrices $\{\mathbf{W}_{\mathcal{F}}^{(l,l-1)}\}$ for predictions.

---

[2]We believe that there is a potential of adapting our techniques to more combinatorial problems beyond TSP.

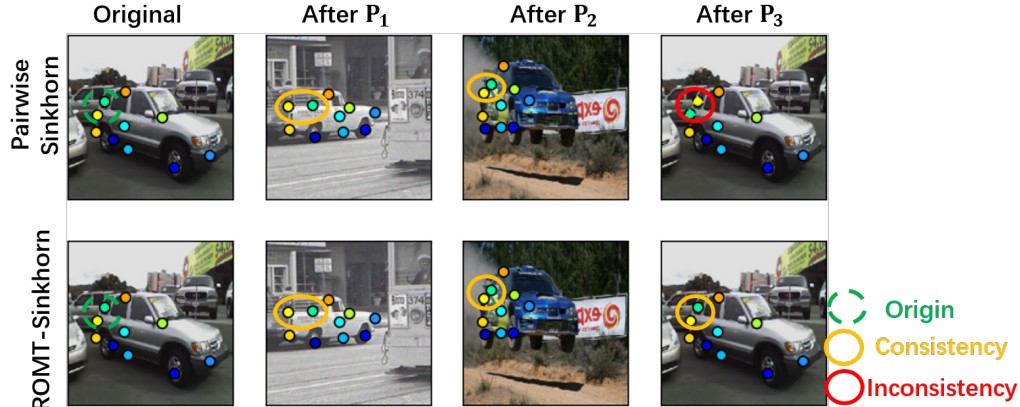

Figure 5: Example point matching results after applying $\mathbf{P}_1$, $\mathbf{P}_2$, and $\mathbf{P}_3$ to the original point set. The features are extracted using NMGM (Wang et al., 2021). We observe that the Sinkhorn method fails to achieve cycle-consistency, while our ROMT-Sinkhorn method successfully maintains cycle-consistency, resulting in the graph after applying $\mathbf{P}_3$ remaining identical to the original graph.

Table 1: **Three Point Sets Matching Comparisons with NMGM (Wang et al., 2021), PCA-GM (Wang et al., 2019), IPCA-GM (Wang et al., 2020b) and CIE-H (Yu et al., 2019) backbone on Willow.** Accuracy (ACC), Consistent rate (CR) and Consistent Accuracy (CACC) (%) are reported and ROMT-Sinkhorn outperforms in CR and CACC and performs competitively in Acc.

| Method | NMGM | | | PCA-GM | | | IPCA-GM | | | CIE-H | | |
|---|---|---|---|---|---|---|---|---|---|---|---|---|
| | ACC | CR | CACC | ACC | CR | CACC | ACC | CR | CACC | ACC | CR | CACC |
| Hungarian (Munkres, 1957) | 91.10 | 93.94 | 84.42 | 56.50 | 59.74 | 33.64 | 40.03 | 63.03 | 22.69 | 37.13 | 68.59 | 18.52 |
| EMD (Dantzig, 1949) | 91.08 | 93.90 | 84.38 | 92.19 | 90.28 | 84.24 | 94.26 | 91.41 | 87.32 | 89.84 | 84.09 | 78.28 |
| Sinkhorn (Cuturi, 2013) | 91.02 | 93.76 | 84.28 | 92.64 | 90.12 | 84.86 | 94.51 | 91.44 | 87.76 | 93.08 | 84.72 | 82.88 |
| MMOT (Elvander et al., 2020) | 91.96 | 93.28 | 87.10 | **93.59** | 94.12 | 89.36 | 94.22 | 90.57 | 87.93 | 94.06 | 86.89 | 85.70 |
| **ROMT-Sinkhorn (ours)** | **92.30** | **99.72** | **88.28** | 93.27 | **98.96** | **90.04** | **95.06** | **95.00** | **90.31** | **94.62** | **98.47** | **91.48** |

Table 2: **Four Point Sets Matching Comparison Results on Willow.**

| Method | NMGM | | | PCA-GM | | | IPCA-GM | | | CIE-H | | |
|---|---|---|---|---|---|---|---|---|---|---|---|---|
| | ACC | CR | CACC | ACC | CR | CACC | ACC | CR | CACC | ACC | CR | CACC |
| Hungarian (Munkres, 1957) | 92.39 | 96.66 | 84.58 | 57.33 | 51.04 | 25.96 | 39.24 | 54.73 | 16.10 | 37.30 | 61.01 | 13.05 |
| EMD (Dantzig, 1949) | 92.39 | 96.66 | 84.38 | 92.47 | 88.12 | 81.44 | 93.83 | 87.75 | 82.21 | 89.85 | 80.24 | 73.18 |
| Sinkhorn (Cuturi, 2013) | 92.28 | 96.26 | 84.40 | 92.74 | 87.04 | 81.34 | 93.81 | 86.93 | 82.14 | 93.08 | 80.45 | 78.31 |
| MMOT (Elvander et al., 2020) | 92.34 | 93.74 | 85.04 | **93.38** | 90.44 | 85.12 | 93.98 | 85.75 | 82.68 | 93.83 | 82.14 | 80.65 |
| **ROMT-Sinkhorn (ours)** | **92.54** | **98.78** | **85.48** | 93.11 | **96.38** | **85.96** | **94.53** | **92.40** | **85.92** | **94.16** | **97.78** | **88.11** |

## 4 EXPERIMENTS

### 4.1 EXPERIMENTS ON VISUAL POINT MATCHING ACROSS SETS

We evaluate on the task of keypoint matching on Pascal VOC dataset with Berkeley annotations (Everingham et al., 2010; Bourdev & Malik, 2009) and Willow Object Class dataset (Cho et al., 2013). In addition to the average accuracy (ACC) (Wang et al., 2021) to evaluate matching results, we develop two metrics called **Consistent Rate (CR)** and **Consistent Accuracy (CACC)** to assess the cycle-consistency effectiveness of the inference method. These metrics are defined as follows:

$$\text{CR} = \frac{1}{n} < \prod_{k=1}^{K} \hat{\mathbf{P}}_k, \mathbf{I}_n > \quad \text{and} \quad \text{CACC} = \frac{1}{n} < \prod_{k=1}^{K} (\hat{\mathbf{P}}_k \odot \mathbf{Y}_k), \mathbf{I}_n > \tag{20}$$

where $\hat{\mathbf{P}}_k$ is the one-hot matching prediction results of $\mathbf{P}_k$ for the $k-$th point set to $(k + 1)-$th set, $\mathbf{Y}_k$ is the ground truth for $\mathbf{P}_k$ and $\odot$ denotes element-wise matrix multiplication and $n$ is the number of points in the first set. Note **CR** refers to the accuracy of forming cycles through matching, while **CACC** represents the accuracy of forming cycles where each feature point within the cycle is matched correctly. We adopt the mean value of CR and CACC as evaluations.

**Results.** The results are summarized in Tab. 1 and Tab. 3. We utilize the previous neural matching models, namely NMGM (Wang et al., 2021), PCA-GM (Wang et al., 2019), IPCA-GM (Wang et al., 2020b) and CIE-H (Yu et al., 2019) ), as the backbone to evaluate our inference algorithm. We compare our ROMT-Sinkhorn algorithm with (Munkres, 1957), EMD (Dantzig, 1949) and Sinkhorn Algorithm (Cuturi, 2013). As shown in Tab. 1, for the Willow dataset, our method outperforms all

Table 3: **Comparison with PCA-GM (Wang et al., 2019), IPCA-GM (Wang et al., 2020b) and CIE-H (Yu et al., 2019) backbone on Pascal VOC dataset with Berkeley annotations.** Accuracy (ACC), Consistent rate (CR) and Consistent Accuracy (CACC) (%) are all reported here and our ROMT-Sinkhorn outperforms in CR and CACC evaluation and performs competitively in Acc.

| Method | PCA-GM | | | IPCA-GM | | | CIE-H | | |
|---|---|---|---|---|---|---|---|---|---|
| | ACC | CR | CACC | ACC | CR | CACC | ACC | CR | CACC |
| Hungarian (Munkres, 1957) | 49.89 | 60.80 | 25.63 | 58.57 | 60.34 | 36.16 | 60.83 | 66.25 | 36.87 |
| EMD (Dantzig, 1949) | 67.03 | 65.59 | 44.44 | 69.07 | 68.09 | 47.24 | 71.60 | 68.67 | 49.87 |
| Sinkhorn (Cuturi, 2013) | 68.14 | 66.64 | 45.28 | 69.83 | 69.02 | 48.94 | 72.67 | 68.54 | 51.04 |
| MMOT (Elvander et al., 2020) | 67.85 | 69.30 | 48.75 | 70.18 | 65.98 | 49.39 | 72.06 | 67.34 | 51.62 |
| **ROMT-Sinkhorn (ours)** | **68.74** | **82.69** | **51.63** | **70.93** | **84.68** | **55.24** | **73.52** | **83.57** | **57.65** |

Table 4: **Results for fusing MLP and VGG11, along with the effect of finetuning the fused models, on MNIST and CIFAR10 respectively.** The numbers below the test accuracies represent the efficiency factor of a fusion technique compared to retaining all the provided models.

| Dataset + Model | $M_A$ | $M_B$ | $M_C$ | Model Fusion without Fine-tuning | | | | Fine-tuning | | |
|---|---|---|---|---|---|---|---|---|---|---|
| | | | | PREDICTION | VANILLA | OTfusion | Ours | VANILLA | OTfusion | Ours |
| MNIST + MLP | 97.72 | 97.75 | 97.69 | 98.00 | 25.59 | 84.16 | 84.32 | 97.88 | 97.96 | **98.00** |
| | | 1× | | 1× | 3× | 3× | 3× | 3× | 3× | 3× |
| CIFAR10 + VGG11 | 90.31 | 90.50 | 90.51 | 91.55 | 9.99 | 88.38 | 88.77 | 90.2 | 89.67 | **90.51** |
| | | 1× | | 1× | 3× | 3× | 3× | 3× | 3× | 3× |

other inference methods by CR and CACC and performs competitively by ACC. For experiments on Pascal VOC with Berkeley annotations, our ROMT-Sinkhorn outperforms others across all backbones. Results for more than three measures are given in Appendix C.

## 4.2 EXPERIMENTS ON NEURAL NETWORK MODEL FUSION

Following the network fusion protocol in (Singh & Jaggi, 2020) using already-trained neural networks for fusion i.e. a single new network whose parameters are fused from the input networks', our experiments are focused on exploring the benefits of fusing multiple models that only differ in their parameter initializations (i.e., seeds). We study this in the context of deep networks such as MLP and VGG11 which have been trained on MNIST and CIFAR10 respectively.

**Results.** We focus on the setting of three model fusions, and the results are presented in Tab. 4. As baselines, we evaluate the performance of prediction ensembling, vanilla averaging, and OTfusion (Singh & Jaggi, 2020), in addition to the individual models. Prediction ensembling involves keeping all the models and averaging their predictions (output layer scores), representing an ideal but unrealistic performance achievable through fusion into a single model. Vanilla averaging refers to directly averaging the model parameters. The numbers below the test accuracies (e.g., 1× and 3×) indicate the factor by which a fusion technique is efficient in maintaining all the given models. We observe that vanilla averaging performs ineffectively in this case and is worse than OTfusion and our proposed method, especially for MLP and VGG11. Our method achieves the best performance. However, both OTfusion and our method do not yet surpass the individual models. As explained in (Singh & Jaggi, 2020), this can be attributed to the combinatorial hardness of the underlying alignment problem and the greedy nature of the algorithm. We also consider fine-tuning from the fused models. Finetuning improves the performance of all three methods: vanilla averaging, OTfusion, and our method. Our method achieves the highest test accuracy, outperforming the other fusion techniques, as shown in Tab. 4. Specifically, when applied to MLP, our method with fine-tuning approaches the performance of the prediction ensemble in terms of test accuracy.

## 5 CONCLUSION AND LIMITATION

We have introduced a generalized form for optimal multiple transportation (OMT), which enables transportation among multiple measures while (softly) preserving cycle-consistent constraints. Building upon the cycle-consistency regularized formulation of OMT, we further propose an iterative Sinkhorn method to approximate the solution. We have applied our OMT algorithms to the domains of visual point set matching and multi-model fusion, both of which have demonstrated competitive performance. We also demonstrate the potential of our approach for solving the challenging TSP problems in probabilistic space, in contrast to traditional methods working in the distance space. For the limitation, our ROMT-Sinkhorn introduces an additional hyperparameter $\delta$ for tuning. Similarly, for the regularized TSP-OMT, the number of hyperparameters increases by $K$.

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

APPENDIX

## A  LAGRANGIAN METHOD FOR REGULARIZED OMT

We first give the derivation of Eq. 12. At first, given the minimization

$$\min_{(\mathbf{P}_k)_k : \mathbf{P}_k \in U(\mathbf{a}^k, \mathbf{a}^{k+1})} \mathcal{L}_1 = \sum_{k=1}^K < \mathbf{C}_k, \mathbf{P}_k > -\epsilon \sum_k H(\mathbf{P}_k) - \delta || \prod_{k=1}^K \tilde{\mathbf{P}}_k - \mathbf{I} ||_F^2, \quad (21)$$

we can adopt the Lagrangian method to solve it. For each coupling $\mathbf{P}_k$, we introduce $(\mathbf{f}_k, \mathbf{g}_k)$ to the constraints in $U(\mathbf{a}^k, \mathbf{a}^{k+1})$, i.e.

$$\mathbf{P}_k \mathbf{1}_{n_{k+1}} = \mathbf{a}^k \quad \text{and} \quad (\mathbf{P}_k)^\top \mathbf{1}_{n_k} = \mathbf{a}^{k+1}, \quad (22)$$

and then we can get the Lagrangian function as

$$\mathcal{L} = \mathcal{L}_1 - \sum_k \big( < \mathbf{f}_k, \mathbf{P}_k \mathbf{1}_{n_{k+1}} - \mathbf{a}^k > + < \mathbf{g}_k, (\mathbf{P}_k)^\top \mathbf{1}_{n_k} - \mathbf{a}^{k+1} > \big) \quad (23)$$

We compute the partial derivative of $\mathcal{L}$ with respect to $\mathbf{P}_k$ as

$$\frac{\partial \mathcal{L}}{\partial \mathbf{P}_k} = \mathbf{C}_k + \epsilon \log \mathbf{P}_k - \mathbf{f}_k \mathbf{1}^\top - \mathbf{1} \mathbf{g}_k^\top - \delta M_k = 0, \quad (24)$$

where $M_k$ is specified as

$$M_k = \frac{\partial f}{\partial \mathbf{P}_k} = \frac{\partial tr(Y^\top Y)}{\partial \mathbf{P}_k} = \frac{\partial || \prod_{k=1}^K \tilde{\mathbf{P}}_k - \mathbf{I} ||_F^2}{\partial \mathbf{P}_k}. \quad (25)$$

Here we set $Y = \prod_{k=1}^K \tilde{\mathbf{P}}_k - \mathbf{I}$ and $f = tr(Y^\top Y)$ in Eq. 25. With the method given in (Hu, 2012), We solve it by utilizing the relationship between matrix derivative and its partial derivatives. Specially, we have

$$df = tr(dY^\top Y) + tr(Y^\top dY) = tr(2Y^\top dY) = tr(\frac{\partial f^\top}{\partial Y} dY), \quad (26)$$

then it is satisfied that $\frac{\partial f^\top}{\partial Y} = 2Y^\top$. For the matrix $\mathbf{P}_k$, we have

$$
\begin{aligned}
df &= tr(\frac{\partial f^\top}{\partial Y} d \prod_{t=1}^K \tilde{\mathbf{P}}_t) = tr(\prod_{t_3=k+1}^K \tilde{\mathbf{P}}_{t_3} \frac{\partial f^\top}{\partial Y} \prod_{t_1=1}^k \tilde{\mathbf{P}}_{t_1} \mathrm{Diag}\left(\frac{1}{\mathbf{a}^k}\right) d\mathbf{P}_k) \\
&= tr(2 \prod_{t_3=k+1}^K \tilde{\mathbf{P}}_{t_3} (\prod_{t_2=1}^K \tilde{\mathbf{P}}_{t_2} - \mathbf{I})^\top \prod_{t_1=1}^k \tilde{\mathbf{P}}_{t_1} \mathrm{Diag}\left(\frac{1}{\mathbf{a}^k}\right) d\mathbf{P}_k) \\
&= tr(\left( 2\mathrm{Diag}\left(\frac{1}{\mathbf{a}^k}\right) (\prod_{t_1=1}^k \tilde{\mathbf{P}}_{t_1})^\top (\prod_{t_2=1}^K \tilde{\mathbf{P}}_{t_2} - \mathbf{I})(\prod_{t_3=k+1}^K \tilde{\mathbf{P}}_{t_3})^\top \right)^\top d\mathbf{P}_k)
\end{aligned} \quad (27)
$$

thus we have

$$M_k = 2\mathrm{Diag}\left(\frac{1}{\mathbf{a}^k}\right) \left(\prod_{t_1=1}^{k-1} \tilde{\mathbf{P}}_{t_1}\right)^\top \left(\prod_{t_2=1}^K \tilde{\mathbf{P}}_{t_2} - \mathbf{I}\right) \left(\prod_{t_3=k+1}^K \tilde{\mathbf{P}}_{t_3}\right)^\top. \quad (28)$$

According to Eq. 24, we have

$$\mathbf{P}_k = \mathrm{Diag}(e^{\mathbf{f}_k/\epsilon}) e^{(-\mathbf{C}_k + \delta M_k)/\epsilon} \mathrm{Diag}(e^{\mathbf{g}_k/\epsilon}) \quad (29)$$

and the iterative Sinkhorn algorithm can be used with the constraints given in Eq. 22.

## B   MORE DETAILS IN TSP-OMT

### B.1   LAGRANGIAN METHOD FOR REGULARIZED TSP-OMT

For the minimization of regularized TSP-OMT, we have

$$\min_{\mathbf{P}} \mathcal{L}_2 = <\mathbf{C}, \mathbf{P}> -\epsilon H(\mathbf{P}) + \sum_k \delta_k ||\mathbf{P}^k - \mathbf{I}||_F^2 \quad \text{s.t.} \quad \mathbf{P}\mathbf{1}_K = \mathbf{1}_K, \, \mathbf{P}^\top \mathbf{1}_K = \mathbf{1}_K. \quad (30)$$

Lagrangian methods are used to solve it here. Introducing the duals $(\mathbf{f}, \mathbf{g})$ to the constraints $\mathbf{P}\mathbf{1}_K = \mathbf{1}_K, \mathbf{P}^\top \mathbf{1}_K = \mathbf{1}_K$, we can get the Lagrangian function

$$\mathcal{L} = \mathcal{L}_2 - <\mathbf{f}, \mathbf{P}\mathbf{1}_K - \mathbf{1}_K> - <\mathbf{g}, \mathbf{P}^\top \mathbf{1}_K - \mathbf{1}_K>. \quad (31)$$

Then we can compute the partial derivative of $\mathcal{L}$ with respect to $\mathbf{P}$ as

$$\frac{\partial \mathcal{L}}{\partial \mathbf{P}} = \mathbf{C} + \epsilon \log \mathbf{P} - \mathbf{f}\mathbf{1}^\top - \mathbf{1}\mathbf{g}^\top + M = 0, \quad (32)$$

where $M$ is specified as

$$M = \sum_k \delta_k \frac{\partial ||\mathbf{P}^k - \mathbf{I}||_F^2}{\partial \mathbf{P}} = \sum_{k=1}^K \sum_{t=0}^{k-1} 2\delta_k (\mathbf{P}^t)^\top (\mathbf{P}^k - \mathbf{I})(\mathbf{P}^{k-1-t})^\top. \quad (33)$$

To prove that, we define $f_k = tr(Y^\top Y) = ||\mathbf{P}^k - \mathbf{I}||_F^2$ where $Y = \mathbf{P}^k - \mathbf{I}$, then

$$df_k = tr\left(2Y^\top dY\right) = tr(\sum_{t=0}^{k-1} \frac{\partial f_k^\top}{\partial Y}(\mathbf{P}^t d\mathbf{P}\mathbf{P}^{k-1-t})) = tr(\sum_{t=0}^{k-1} \mathbf{P}^{k-1-t} \frac{\partial f_k^\top}{\partial Y} \mathbf{P}^t dP)$$
$$= tr\left(\left(2\sum_{t=0}^{k-1}(\mathbf{P}^t)^\top (\mathbf{P}^k - \mathbf{I})(\mathbf{P}^{k-1-t})^\top\right)^\top dP\right), \quad (34)$$

Thus we have

$$\frac{\partial f_k}{\partial P} = \sum_{t=0}^{k-1} 2(\mathbf{P}^t)^\top (\mathbf{P}^k - \mathbf{I})(\mathbf{P}^{k-1-t})^\top. \quad (35)$$

So we can get that

$$M = \sum_k \delta_k \frac{\partial f_k}{\partial \mathbf{P}} = \sum_{k=1}^K \sum_{t=0}^{k-1} 2\delta_k (\mathbf{P}^t)^\top (\mathbf{P}^k - \mathbf{I})(\mathbf{P}^{k-1-t})^\top. \quad (36)$$

According to Eq. 32, we can the solution form

$$\mathbf{P} = \text{Diag}(e^{\mathbf{f}/\epsilon}) e^{-(\mathbf{C}+M)/\epsilon} \text{Diag}(e^{\mathbf{g}/\epsilon}). \quad (37)$$

Thus the iterative Sinkhorn algorithm can be applied for calculation.

### B.2   GREEDY METHOD FOR SEARCHING WITH PROBABILITY MATRIX

With a known probability matrix calculated by Sinkhorn or Algorithm 2, we can apply the Algorithm 3 to get the TSP path.

Table 5: **Experiments of four measures: Comparison with PCA-GM (Wang et al., 2019), IPCA-GM (Wang et al., 2020b) and CIE-H (Yu et al., 2019) backbone on Pascal VOC dataset with Berkeley annotations.** Accuracy (ACC), Consistent rate (CR) and Consistent Accuracy (CACC) (%) are all reported here and our ROMT-Sinkhorn outperforms in CR and CACC evaluation and performs competitively in Acc.

| Method | PCA-GM | | | IPCA-GM | | | CIE-H | | |
|---|---|---|---|---|---|---|---|---|---|
| | ACC | CR | CACC | ACC | CR | CACC | ACC | CR | CACC |
| Hungarian (Munkres, 1957) | 54.30 | 55.96 | 22.01 | 62.65 | 57.08 | 34.01 | 65.86 | 58.23 | 34.64 |
| EMD (Dantzig, 1949) | 70.45 | 61.06 | 40.04 | 70.71 | 62.90 | 42.28 | 73.91 | 64.36 | 45.04 |
| Sinkhorn (Cuturi, 2013) | 70.49 | 61.31 | 40.53 | 71.33 | 63.92 | 43.92 | 74.53 | 63.74 | 45.78 |
| MMOT (Elvander et al., 2020) | 70.56 | 61.41 | **41.73** | 71.32 | 61.44 | 43.93 | 74.69 | 63.19 | 46.17 |
| **ROMT-Sinkhorn (ours)** | 70.60 | **63.74** | 41.54 | **71.86** | **78.99** | **50.01** | **75.40** | **76.41** | **51.60** |

---

**Algorithm 3** Greedy

**Input:** the coupling $\mathbf{P}$
**Output:** the path $tour$
  Initialize $tour = []$
  $i, j = \text{where}(\mathbf{P} == \mathbf{P}.max())$
  $\mathbf{P}[i, :] = \mathbf{P}[:, j] = \mathbf{P}[:, i] = 0$
  $k = j$
  $tour.append(i)$
  $tour.append(j)$
  **for** $m = 1, \ldots, n - 2$ **do**
    $i, j = k, \text{where}(\mathbf{P}[\mathbf{k}, :] == \mathbf{P}[\mathbf{k}, :].max())$
    $\mathbf{P}[i, :] = \mathbf{P}[:, j] = 0$
    $k = j$
    $tour.append(j)$
  **end for**

---

## C MORE EXPERIMENTAL RESULTS

The experimental results of four measures are given in Tab. 2 and Tab. 5 for visual matching and Tab. 6 for model fusion. Fig. 6 is the convergence cure for visual matching for Willow_3GM data with IPCA as the backbone, which shows that our ROMT-Sinkhorn can easily converge with several epochs. Note the error is set as the L2 norm for the iterative coupling difference (i.e. $\sum_k ||\mathbf{P}_k^t - \mathbf{P}_k^{t-1}||_2$ with current iterative number $t$).

Runtimes of the pair-wise Sinkhorn and ROMT-Sinkhorn are shown in Tab. 7. The time complexity of ROMT-Sinkhorn is $L$ times that of the pair-wise Sinkhorn, where $L$ represents the number of iterations.

For the order of sets, we first consider three measure case with $A$, $B$, and $C$, with a mapping order defined as $A \to B$, $B \to C$, and $C \to A$, represented by matching matrices $\mathbf{P}_1$, $\mathbf{P}_2$, and $\mathbf{P}_3$. Consistency requires that $\mathbf{P}_1 \mathbf{P}_2 \mathbf{P}_3 = \mathbf{I}$. If we switch the order to $A \to C$, $C \to B$, and $B \to A$, with matching matrices $\mathbf{P}_3^\top$, $\mathbf{P}_2^\top$, and $\mathbf{P}_1^\top$, the consistency becomes $\mathbf{P}_3^\top \mathbf{P}_2^\top \mathbf{P}_1^\top = \mathbf{P}_1 \mathbf{P}_2 \mathbf{P}_3 = \mathbf{I}$. Therefore, for the case of three measures ($K = 3$), switching the order (of Fig. 5) does not affect the problem formulation. We test the experiments on Willow_3GM with IPCA-GM as backbone and the results are given in Tab. 8. Although the problems with and without switching the order are equivalent, the results may differ slightly due to initialization and other factors.

As for $K > 3$, switching the order does indeed impact the formulation of the problem. However, note our directed cyclical structure is essentially a subgraph of pairwise structure. When the latter is satisfied, the former is also satisfied, allowing our method to still improve the matching performance. We show the experiments of switching the second and third set order in Tab. 9.

We conducted an ablation study for visual matching experiments by varying $\delta$ and $\epsilon$. The results are given in Tab. 10.

Table 6: **Comparison results for four model fusion.**

| Dataset + Model | $M_A$ | $M_B$ | $M_C$ | $M_D$ | Model Fusion without Fine-tuning | | | | Fine-tuning | | |
|---|---|---|---|---|---|---|---|---|---|---|---|
| | | | | | PREDICTION | VANILLA | OTfusion | Ours | VANILLA | OTfusion | Ours |
| MNIST + MLP | 97.72 | 97.75 | 97.69 | 97.26 | 97.91 | 20.12 | 86.52 | 87.13 | 97.86 | 98.06 | **98.08** |
| | | 1× | | | 1× | 4× | 4× | 4× | 4× | 4× | 4× |
| CIFAR10 + VGG11 | 90.31 | 90.50 | 90.51 | 90.58 | 91.79 | 10.02 | 88.66 | 88.94 | 10.86 | 89.79 | **90.35** |
| | | 1× | | | 1× | 4× | 4× | 4× | 4× | 4× | 4× |

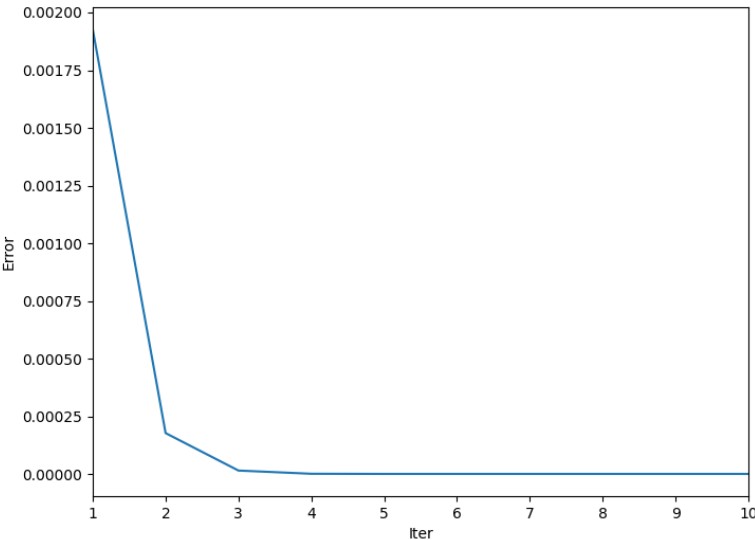

Figure 6: The convergence curve of ROMT-Sinkhorn for Willow_3GM data with IPCA backbone.

Table 7: **Runtimes of pair-wise Sinkhorn and ROMT-Sinkhorn in different experiments.** We set $L = 5$ and get the average runtime of pair-wise Sinkhorn and ROMT-Sinkhorn on all samples of an experiment.

| Experiment | runtime of pair-wise Sinkhorn(s) | runtime of ROMT-Sinkhorn(s) |
|---|---|---|
| Willow_3GM | 0.0113 | 0.0583 |
| Willow_4GM | 0.0150 | 0.0782 |
| Willow_5GM | 0.0188 | 0.0989 |

Table 8: **The results for the case of three measures (K=3 ) when switching the order on Willow_3GM with IPCA-GM as backbone.**

| Willow_3GM with IPCA-GM | ACC | CACC | CR |
|---|---|---|---|
| ROMT-Sinkhorn : A→B, B→C, C→A | 0.9506 | 0.9031 | 0.9500 |
| ROMT-Sinkhorn : A→C, C→B, B→A | 0.9459 | 0.8974 | 0.9477 |

Table 9: **The results for the case of three measures (K=4 ) when switching the second and third set order on Willow_4GM with IPCA-GM as backbone.**

| Willow_4GM with IPCA-GM | ACC | CACC | CR |
|---|---|---|---|
| ROMT-Sinkhorn without switching | 0.9453 | 0.8592 | 0.9240 |
| ROMT-Sinkhorn with switching the order | 0.9460 | 0.8557 | 0.9222 |

Table 10: **Ablation study for visual matching experiments by varying $\delta$ and $\epsilon$.**

| $\delta$ | $\epsilon$ | ACC | CACC | CR |
|---|---|---|---|---|
| 0.001 | 1e-9 | 0.9412 | 0.8767 | 0.9158 |
| 0.001 | 1e-10 | 0.9412 | 0.8767 | 0.9158 |
| 0.01 | 1e-9 | 0.9442 | 0.8967 | 0.9475 |
| 0.001 | 1e-11 | 0.9412 | 0.8767 | 0.9158 |
| 0.01 | 1e-10 | 0.9442 | 0.8967 | 0.9475 |
| 0.01 | 1e-11 | 0.9442 | 0.8967 | 0.9475 |
| 0.1 | 1e-9 | 0.9382 | 0.9087 | 0.9951 |
| 0.1 | 1e-10 | 0.9382 | 0.9087 | 0.9951 |
| 0.1 | 1e-11 | 0.9382 | 0.9087 | 0.9951 |

