# OpenReview forum: "Optimal Multiple Transport with Applications to Visual Matching, Model Fusion and Beyond"
_ICLR.cc/2024/Conference — Submitted to ICLR 2024_

### Official Review · Reviewer_aQyc · 2023-10-19

**Soundness:** 3 good
**Presentation:** 3 good
**Contribution:** 3 good
**Rating:** 5
**Confidence:** 4

**Summary:**

This paper proposes Optimal Multiple Transport (OMT), which handles the case where one needs to compute multiple optimal transports
between several distributions that are linked together by a cyclic dependency structure (a sample from the first distribution should ‘travel’
back to its original location after being transported to several other distributions). They propose an algorithm to solve an associated optimization problem,  which is an extension of the Sinkhorn algorithm. After making links with the traveling salesman problem (TSP), they present two applications of their algorithm to visual multi-point matching and multi model fusion, where the proposed approach leads to benefits in terms of performance.

**Strengths:**

- The idea of enforcing consistency between multiple transport is original, neat and interesting.
 - the transposition to the TSP problem is very interesting.

**Weaknesses:**

- it is not clear if in the general setting there exist a solution to problem (8). What are the conditions for the feasible set not to be empty ?
- lack of justifications and analyze for algorithm 1.
- how to build a cycle (ordering of measures) is not discussed in the paper

(see my comments below)

**Questions:**

In general, I like the idea of cycle consistency argued by the paper, and the connection to TSP is very stimulating.. However, and this is preventing me from giving an higher score, there are a number of issues with this paper. Provided that some of them are solved during the discussion phase, I will be willing to change my evaluation.  Here are potential questions and remarks for the authors :

- Existence of solution for Eq 9 should be discussed. Notably, it seems that as soon as there is mass splitting or whenever one measure has a small number of atoms, the composition (product) of couplings can not yield identity (as soon as there might be rank deficiency in the coupling matrix), and the feasible set is empty. In the reviewer’s opinion, this should be detailed or analyzed.

- Regarding the optimization part, there is no clear justification for the alternating scheme proposed here (solving sinkhorns, then updating the cost matrix for every sub-problems). I think there might be connections to Generalized Conditional Gradients or Majoration-Minimization methods here (as the ones used in the POT library). I encourage the authors to better justify their algorithm 1. Also, what is the impact of the entropy parameter $\epsilon$ in solving the problem (10) ? How do you set it ? How many iterations wrt. K are needed for finding a solution ? In the reviewer’s opinion, this would deserve a discussion in the paper.

- Authors do not mention the case when K=2. However, I believe that it might also be interesting. In the case where the number of samples (atoms) is the same in the two distributions (with uniform distributions), we have obviously that $P P^T = Id$ but when  mass splitting occur,  it is not always the case. What is then the impact of the regularization on the original sinkhorn problem ?

- In the applications of OMT (visual point matching, or neural model fusion), how do you define the sequences/orderings of measures ? Are there any change in the performances if one changes this ordering (as we can expect because of the non-commutativity of coupling matrices product) ?  More generally, and as soon as K>2, I guess that the choice of the cycle (when given a set of input measures) is in itself a problem. It could be better discussed in the paper

Minor comments
 - in Eq 7 the first k should be a capital K
- I do not really understand the difference between Eq. 10 and 17
- Figure 3 could be enhanced to better distinguish the differences between the two versions of the transport plans, which is difficult to see at a first glance
- p6, what do you mean by ‘… Algorithm 2 can not achieve the ideal closed-loop solution which may be
due to the simple setting of δk and too many regularized terms of closed-loop constraints’ ?
- in Multi point matching, performance measures such as CR or CACC show a bias toward the presented method, since it directly relates to the criterion which is optimized.

---

> ### Author Response · Authors · 2023-11-19
>
> **** Q1. how to build a cycle (ordering of measures for $K>2$) is not discussed in the paper
>
> **** A1. For $K=3$, consider three sets: $A$, $B$, and $C$, with a mapping order defined as $A\to B$, $B\to C$, and $C\to A$, represented by matching matrices $\mathbf{P}_1$, $\mathbf{P}_2$, and $\mathbf{P}_3$. Consistency requires that $\mathbf{P}_1\mathbf{P}_2\mathbf{P}_3=\mathbf{I}$. If we switch the order to $A\to C$, $C\to B$, and $B\to A$, with matching matrices $\mathbf{P}_3^\top$, $\mathbf{P}_2^\top$, and $\mathbf{P}_1^\top$, the consistency becomes $\mathbf{P}_3^\top\mathbf{P}_2^\top\mathbf{P}_1^\top=\mathbf{P}_1\mathbf{P}_2\mathbf{P}_3=\mathbf{I}$. Therefore, for the case of three measures ($K=3$), switching the order (of Fig. 5) does not affect the problem formulation.We test the experiments on Willow_3GM with IPCA-GM as backbone and the results are given as follows. Although the problems with and without switching the order are equivalent, the results may differ slightly due to initialization and other factors.
>
> | Willow_3GM with IPCA-GM       |  ACC    | CACC   | CR     |
> | ------------------------------|------ | ------ | ------ |
> | ROMT-Sinkhorn : A→B, B→C, C→A | 0.9506 | 0.9031 | 0.9500 |
> | ROMT-Sinkhorn : A→C, C→B, B→A | 0.9459 | 0.8974 | 0.9477 |
>
> As for $K>3$, switching the order does indeed impact the formulation of the problem. However, note our directed cyclical structure is essentially a subgraph of pairwise structure. When the latter is satisfied, the former is also satisfied, allowing our method to still improve the matching performance. We show the experiments of switching the **second** and **third** set order as follows:
>
> | Willow_4GM with  IPCA-GM               | ACC    | CACC   | CR     |
> | -------------------------------------- | ------ | ------ | ------ |
> | ROMT-Sinkhorn without switching        | 0.9453 | 0.8592 | 0.9240 |
> | ROMT-Sinkhorn with switching the order | 0.9460 | 0.8557 | 0.9222 |
>
> It can be observed that the results with and without switching the order are actually very close to each other.
>
> **** Q2.  Justification for the alternating scheme. I think there might be connections to Generalized Conditional Gradients or Majoration-Minimization methods here.
>
> **** A2. We initially proposed ROMT-Sinkhorn by referencing the computation and derivation of the Entropic Gromov-Wasserstein Distance. We are grateful to the reviewer for reminding us to understand our ROMT-Sinkhorn from the perspective of optimization algorithm frameworks. While we believe that ROMT-Sinkhorn does not seem to have a direct connection with Generalized Conditional Gradients or Majoration-Minimization, we have discovered that it can be associated with the **Alternating Direction Method of Multipliers (ADMM)**. The outer loop of our algorithm essentially optimizes different variables $\{\mathbf{P}_k\}_k$, while the inner Sinkhorn iterative loop optimizes multipliers ${(f_k,g_k)}_k$. Once again, we appreciate the reviewer's reminder, which has provided further optimization potential for ROMT-Sinkhorn.
>
> **** Q3. The existence to problem (8)
>
> **** A3. Thanks to the reviewer for pointing out the limitation in the existence of problem (8) theoretically. The existence problem is both important and challenging, and currently, we are unable to provide proof of its existence. However, we firmly believe that in the matching setting (i.e., $\mathbf{a}^k=\mathbf{1}_n$), a solution to the problem (8) does exist.
>
> In the general case, it seems to be related to the probability vectors ${\mathbf{a}^k}$ and the dimension numbers ${n_k}$. From the perspective of degrees of freedom, for any given $P_k$, assuming $\{P_s\}_{s\neq k}$,
>
> there exist $n_k\times n_{k+1}$ variables and $n_k+n_{k+1}-1+n_1^2$ equations. This suggests that when $n_k>>n_1$ and $n_{k+1}>>n_1$, the equations have infinitely many solutions.
>
> **** Q4. $K=2$ case. I believe that it might also be interesting. In the case where the number of samples (atoms) is the same in the two distributions (with uniform distributions), we have obviously that $PP^T=I$. When mass splitting occur, it is not always the case. What is then the impact of the regularization on the original sinkhorn problem ?
>
> **** A4. We appreciate the reviewer for bringing up an interesting special case. In fact, for $K=2$, the consistency condition is not $PP^T=I$, but rather $diag(1/a)PP^Tdiag(1/b)=I$, where $a$ and $b$ are marginals. Additionally, when using entropic regularization for coupling, the consistency to $I$ is no longer satisfied because the coupling matrix is no longer sparse. This leads to the product of these matrices also being non-sparse.

---

> > ### Author Response · Authors · 2023-11-19
> >
> > **** Q4. The impact of $\epsilon$ and how many iterations.
> >
> > **** A4. We conducted an ablation study for visual matching experiments by varying $\delta$ and $\epsilon$. The results are given as follows.
> >
> > **Willow_3GM： IPCA-GM-ROMT-Sinkhorn**
> > | $\delta$ | $\epsilon$ | ACC    | CACC   | CR     |
> > | -------- | ---------- | ------ | ------ | ------ |
> > | 0.001    | 1e-9       | 0.9412 | 0.8767 | 0.9158 |
> > | 0.001    | 1e-10      | 0.9412 | 0.8767 | 0.9158 |
> > | 0.01     | 1e-9       | 0.9442 | 0.8967 | 0.9475 |
> > | 0.001    | 1e-11      | 0.9412 | 0.8767 | 0.9158 |
> > | 0.01     | 1e-10      | 0.9442 | 0.8967 | 0.9475 |
> > | 0.01     | 1e-11      | 0.9442 | 0.8967 | 0.9475 |
> > | 0.1      | 1e-9       | 0.9382 | 0.9087 | 0.9951 |
> > | 0.1      | 1e-10      | 0.9382 | 0.9087 | 0.9951 |
> > | 0.1      | 1e-11      | 0.9382 | 0.9087 | 0.9951 |
> >
> > We find $\epsilon$ affect less when it is small enough. As for $\delta$, it seems to be a tradeoff between ACC and CACC/CR. Regarding the iteration numbers, we found that convergence was achieved when $L=5$ (see Fig. 6 in new version).
> >
> >
> > **** Q5. what do you mean by ‘… Algorithm 2 can not achieve the ideal closed-loop solution which may be due to the simple setting of δk and too many regularized terms of closed-loop constraints’ ?
> >
> > **** A5. The adjustment of $\delta_k$ is necessary because $(P^k)_{ii}$ can be the largest value among the row and column for some $k$, while it may not be the case for others. Simply setting $\delta_k$ as a constant for all $k$ is not advisable at the moment. It seems that a feedback function needs to be defined based on the results of $P^k$ to determine the appropriate values of $\delta_k$.
> >
> > **** Q6. in Multi point matching, performance measures such as CR or CACC show a bias toward the presented method, since it directly relates to the criterion which is optimized.
> >
> > **** A6. 1) We believe that the relationship between CACC and consistency regularization is not direct because CACC requires label/ground truth verification, while the latter focuses on improving cycle-consistency, which is independent of the correctness of the matching. 2) Both CR and CACC have meaningful and interpretable implications for multiple matching, which can be regarded as extensions and refinements of the ACC metric. We can redefine ACC from a matrix computation perspective as follows:
> > $$
> >    ACC = \frac{1}{n} <\prod_{k=1}^2(\hat{\mathbf{P}}_k\odot \mathbf{Y}_k),\mathbf{I}_n>
> > $$
> > where $\mathbf{Y}_k$ represents the ground truth as a one-hot matrix. $\hat{\mathbf{P}}_1$ is the probability matrix for matching from the first set of points to the second set, and $\hat{\mathbf{P}}_2$ is the probability matrix for matching from the second set to the first set. In this particular case, we have $\hat{\mathbf{P}}_1= \hat{\mathbf{P}}_2^T$ and $\mathbf{Y}_1=\mathbf{Y}_2^T$. From this perspective, we define the CR and CACC metrics, incorporating the concept of K-marginal transportation:
> >
> > $$
> >     \text{CR} = \frac{1}{n}<\prod_{k=1}^K\hat{\mathbf{P}}_k,\mathbf{I}_n> \text{ and } \text{CACC}=\frac{1}{n}<\prod_{k=1}^K(\hat{\mathbf{P}}_k\odot \mathbf{Y}_k),\mathbf{I}_n>
> > $$
> > Note that $CR$ refers to the accuracy of forming cycles through matching, capturing the ability to establish correct pairwise matches within the set of $K$ points. On the other hand, CACC represents the accuracy of correctly matching all $K$ points, taking into account the entire cycle instead of only considering pairs of points as in traditional ACC metrics.

---

> ### Author Response · Authors · 2023-11-22
>
> Dear Reviewer,
>
> Thank you very much for taking the time to review our paper. We appreciate the insightful comments and suggestions you provided. As the deadline for the discussion phase approaches, we have carefully considered your feedback and prepared a response to address the concerns raised. We kindly request your feedback on our response and welcome any further questions or suggestions you may have.
>
> Once again, we would like to express our gratitude for your time and attention. Your expertise and guidance are invaluable to us in refining our research.
>
> Best regards,
>
> The authors

---

> > ### Comment · Reviewer_aQyc · 2023-11-22
> > **Some questions remain**
> >
> > Dear Authors,
> >
> > Thank you for adressing my concerns and your detailed review. I appreciate the efforts put into this process.
> > I think some issues remain:
> >  - in answer to Q1, you highlight the fact that the consistencies are equivalent with respect to the order in which you go through the sets A, B and C. This is a nice observation, but this is only true at convergence, and if and only if Identity can be reached by the optimization algorithm, and if such a solution exists and is in the feasible set;
> >  - In answer to Q2, I believe the connection to ADMM could be described in the paper;
> >  - for Q3 I understand that the problem of establishing a formal existence of non-emptiness of the solution set is difficult for the general case. I still think that this is a major issue, as far as it is not clear what the proposed regularization is actually achieving in this case. A solution (less general though) might be to solely frame the problem in the context of assignment (same number of atoms + uniform distribution for all the measures) .
> >
> > I will keep my initial score for this submission. I still believe that the content is valuable and the idea of cycle consistency is neat and interesting, but I think that the paper is not ready for publication

---

### Official Review · Reviewer_uVEA · 2023-11-01

**Soundness:** 2 fair
**Presentation:** 3 good
**Contribution:** 2 fair
**Rating:** 5
**Confidence:** 3

**Summary:**

This paper proposes a generalized form, called optimal multiple transportation (OMT), for optimal transport with multiple probability measures.
In particular, given K distribution measures and their samples, the authors organize them as a loop and consider the pairwise OT problems with a cycle-consistency regularizer. As a special case of this problem, the authors further consider applying the method to solve TSP, in which multiple push-forward steps are implemented by the multiplication of the OT plan and itself and regularized by the cycle-consistency regularizer.
A Sinkhorn-like algorithm is proposed to solve the OMT problem approximately.
The proposed OMT is shown to be effective according to empirical results in the domains of visual point set matching and multi-model fusion.
The OMT also demonstrates the potential for solving the challenging TSP problems in probabilistic space, in contrast to traditional methods working in the distance space.

**Strengths:**

1. This paper is well-organized and well-written. The authors provide sufficient details about their work and easy to understand.

2. Generalizing the optimal transport model to multiple distribution measures is an interesting and significant problem.

3. It’s interesting to see the application of OMT on the TSP problem.

**Weaknesses:**

1. I don’t find much novelty in the idea of minimizing the transportation costs between each pair of distributions and using the constraint $\prod^K_{k=1} \tilde{P}_k=I$  to ensure cycle-consistency. It’s also quite common to apply entropy regularization and transform the cycle-consistency into a regularizer.

2. What about the runtime efficiency of OMT? It would be nice if the authors could show the convergence curve of the proposed algorithm.

3. What about the stability of OMT? Given K sample sets/distributions, the method requires organizing the sets as a loop. Is it robust to the order of the sets? An analytic experiment should be added.

4. The baselines shown in Table 1 are relatively weak. Why don’t the authors consider multi-marginal transport as a baseline? In particular, in the sample/distribution matching tasks, many OT-based methods can be used to achieve multi-source matching, e.g., Wasserstein barycenter [1], Multi-marginal OT (MMOT) [2], and so on.

5. I suggest the authors add a stronger ablation study about the effectiveness of the cycle-consistency regularizer. More baselines[3] should be considered in the experiment of Fig.5.

[1] Cuturi, Marco, and Arnaud Doucet. "Fast computation of Wasserstein barycenters." International conference on machine learning. PMLR, 2014.

[2] Elvander, Filip, et al. "Multi-marginal optimal transport using partial information with applications in robust localization and sensor fusion." Signal Processing 171 (2020): 107474.

[3] Yu, Tianshu, Runzhong Wang, Junchi Yan, and Baoxin Li. "Learning deep graph matching with channel-independent embedding and hungarian attention." In International conference on learning representations. 2019.

**Questions:**

Please see above.

---

> ### Author Response · Authors · 2023-11-19
>
> #### Q1: I don’t find much novelty in the idea of minimizing the transportation costs between each pair of distributions and using the constraint $\prod_k \tilde{P}_k=I$. It’s also quite common to apply entropy regularization and transform the cycle-consistency into a regularizer.
>
> #### A1: We have to feel sorry that you (and perhaps also other readers) think the constraint $\prod_k \tilde{P}_k=I$ for OMT not sufficiently novel, which is worth our further expaination. Specifically, we would like to clarify the following points: 1) Adding this constraint to OT is not an easy task for solving, and adopting the F-norm for relaxation might be the most straightforward computational approach. We have also experimented with alternative norms/divergences, all of which failed with the iterative Sinkhorn method. 2) Our main objective is to achieve consistency among multiple OT solutions, and the mentioned constraint is just one way of achieving this goal. Previous works have not explored this aspect. 3）Transportation consistency has strong implications. In addition to the connection with the Traveling Salesman Problem (TSP) mentioned in the paper, we believe it is also related to the Dynamic Formulation of OT, which describes the OT between two measures as a curve linking with fluid dynamics.
>
> #### Q2: What about the runtime efficiency of OMT? It would be nice if the authors could show the convergence curve of the proposed algorithm.
>
> #### A2: The time complexity of ROMT-Sinkhorn is $L$ times that of the pair-wise Sinkhorn, where $L$ represents the number of iterations. The convergence curve is presented in the new paper version in the Fig. 6 in Appendix, where the error is defined as the mean of the L2 norm between the couplings of two iterations. As for the runtime,  the results are given as follows:
>
> | Experiment | runtime of pair-wise Sinkhorn(s) | runtime of ROMT-Sinkhorn(s) |
> | ---------- | -------------------------------- | --------------------------- |
> | Willow_3GM | 0.0113                           | 0.0583                      |
> | Willow_4GM | 0.0150                           | 0.0782                      |
> | Willow_5GM | 0.0188                           | 0.0989                      |
>
> #### Q3: What about the stability of OMT? Given K sample sets/distributions, the method requires organizing the sets as a loop. Is it robust to the order of the sets? An analytic experiment should be added.
>
> #### A3: About the order of sets, we first consider three measure case with $A$, $B$, and $C$, with a mapping order defined as $A\to B$, $B\to C$, and $C\to A$, represented by matching matrices $\mathbf{P}_1$, $\mathbf{P}_2$, and $\mathbf{P}_3$. Consistency requires that $\mathbf{P}_1\mathbf{P}_2\mathbf{P}_3=\mathbf{I}$. If we switch the order to $A\to C$, $C\to B$, and $B\to A$, with matching matrices $\mathbf{P}_3^T$, $\mathbf{P}_2^T$, and $\mathbf{P}_1^\top$, the consistency becomes $\mathbf{P}_3^\top\mathbf{P}_2^\top\mathbf{P}_1^\top=\mathbf{P}_1\mathbf{P}_2\mathbf{P}_3=\mathbf{I}$. Therefore, for the case of three measures ($K=3$), switching the order (of Fig. 5) does not affect the problem formulation. We test the experiments on Willow_3GM with IPCA-GM as backbone and the results are given as follows. Although the problems with and without switching the order are equivalent, the results may differ slightly due to initialization and other factors.
>
> | Willow_3GM with IPCA-GM       |  ACC    | CACC   | CR     |
> | ------------------------------|------ | ------ | ------ |
> | ROMT-Sinkhorn : A→B, B→C, C→A | 0.9506 | 0.9031 | 0.9500 |
> | ROMT-Sinkhorn : A→C, C→B, B→A | 0.9459 | 0.8974 | 0.9477 |
>
>
>
> As for $K>3$, switching the order does indeed impact the formulation of the problem. However, note our directed cyclical structure is essentially a subgraph of pairwise structure. When the latter is satisfied, the former is also satisfied, allowing our method to still improve the matching performance. We show the experiments of switching the **second** and **third** set order as follows:
>
> | Willow_4GM with  IPCA-GM               | ACC    | CACC   | CR     |
> | -------------------------------------- | ------ | ------ | ------ |
> | ROMT-Sinkhorn without switching        | 0.9453 | 0.8592 | 0.9240 |
> | ROMT-Sinkhorn with switching the order | 0.9460 | 0.8557 | 0.9222 |
>
> It can be observed that the results with and without switching the order are actually very close to each other.

---

> > ### Author Response · Authors · 2023-11-19
> >
> > #### Q4: The baselines shown in Table 1 are relatively weak. Why don’t the authors consider multi-marginal transport as a baseline? In particular, in the sample/distribution matching tasks, many OT-based methods can be used to achieve multi-source matching, e.g., Wasserstein barycenter, Multi-marginal OT (MMOT), and so on.
> >
> > #### A4: We include Multi-Marginal OT as baselines in the visual matching experiments, where the cost tensor is defined as $$C_{ijk}=C'_{ij}+C''_{jk}+C'''_{ki}$$, with cost matrices $C', C'', C'''$.
> > The Bregman iterative method [2] is employed for calculation. Following your advice, we utilize $\sum_k P_{ijk}$ as the probability for the transportation between $i$ and $j$, and similarly for $\sum_j P_{ijk}$ and $\sum_i P_{ijk}$. The experimental results are presented as follows:
> >
> > | Willow_3GM                   | ACC        | CACC       | CR         |
> > | ---------------------------- | ---------- | ---------- | ---------- |
> > | IPCA-GM-Hungarian            | 0.4003     | 0.2269     | 0.6303     |
> > | IPCA-GM-EMD                  | 0.9426     | 0.8732     | 0.9141     |
> > | IPCA-GM-Sinkhorn             | 0.9451     | 0.8776     | 0.9144     |
> > | IPCA-GM-MMOT                 | 0.9422     | 0.8793     | 0.9057     |
> > | IPCA-GM-ROMT-Sinkhorn (ours) | **0.9506** | **0.9031** | **0.9500** |
> >
> > | Willow_4GM                   | ACC        | CACC       | CR         |
> > | ---------------------------- | ---------- | ---------- | ---------- |
> > | IPCA-GM-Hungarian            | 0.3924     | 0.1610     | 0.5473     |
> > | IPCA-GM-EMD                  | 0.9383     | 0.8221     | 0.8775     |
> > | IPCA-GM-Sinkhorn             | 0.9381     | 0.8214     | 0.8693     |
> > | IPCA-GM-MMOT                 | 0.9398     | 0.8268     | 0.8575     |
> > | IPCA-GM-ROMT-Sinkhorn (ours) | **0.9453** | **0.8592** | **0.9240** |
> >
> > Besides, for the Wasserstein barycenter-based methods, we are unsure about how to set the barycenter to obtain matching/transportation results for the samples. We would greatly appreciate it if you could provide some advice on this matter.
> >
> >
> >
> > #### Q5: I suggest the authors add a stronger ablation study about the effectiveness of the cycle-consistency regularizer. More baselines[1] should be considered in the experiment of Fig.5
> >
> > #### A5: We conducted an ablation study by varying $\delta$ and $\epsilon$, and the results are shown as follows. Additionally, the channel-independent embedding and Hungarian attention proposed in [1] essentially modify the network or loss to improve representation, which can only serve as our backbone rather than a baseline. Fig. 5 essentially visualizes how RMOT-Sinkhorn corrects vanilla Sinkhorn and the significance of replacing the feature extractor is not substantial.
> >
> > **Willow_3GM： IPCA-GM-ROMT-Sinkhorn**
> > | $\delta$ | $\epsilon$ | ACC    | CACC   | CR     |
> > | -------- | ---------- | ------ | ------ | ------ |
> > | 0.001    | 1e-9       | 0.9412 | 0.8767 | 0.9158 |
> > | 0.001    | 1e-10      | 0.9412 | 0.8767 | 0.9158 |
> > | 0.001    | 1e-11      | 0.9412 | 0.8767 | 0.9158 |
> > | 0.01     | 1e-9       | 0.9442 | 0.8967 | 0.9475 |
> > | 0.01     | 1e-10      | 0.9442 | 0.8967 | 0.9475 |
> > | 0.01     | 1e-11      | 0.9442 | 0.8967 | 0.9475 |
> > | 0.1      | 1e-9       | 0.9382 | 0.9087 | 0.9951 |
> > | 0.1      | 1e-10      | 0.9382 | 0.9087 | 0.9951 |
> > | 0.1      | 1e-11      | 0.9382 | 0.9087 | 0.9951 |
> >
> > [1] Yu, Tianshu, Runzhong Wang, Junchi Yan, and Baoxin Li. "Learning deep graph matching with channel-independent embedding and hungarian attention." In International conference on learning representations. 2019.
> >
> > [2] Benamou, Jean-David, Guillaume Carlier, Marco Cuturi, Luca Nenna, and Gabriel Peyré. "Iterative Bregman projections for regularized transportation problems." SIAM Journal on Scientific Computing 37, no. 2 (2015): A1111-A1138.

---

> ### Author Response · Authors · 2023-11-22
>
> Dear Reviewer,
>
> Thank you for taking the time to review our paper. We greatly appreciate your insightful comments and suggestions, which have helped us improve the quality of our work. As the deadline for the discussion phase approaches, we have carefully considered your feedback and prepared a response to address the concerns raised. We kindly request your feedback on our response and welcome any further questions or suggestions you may have.
>
> Once again, we would like to express our gratitude for your time and attention. Your expertise and guidance are invaluable to us in refining our research.
>
> Best regards,
>
> The authors

---

### Official Review · Reviewer_16yx · 2023-11-01

**Soundness:** 3 good
**Presentation:** 3 good
**Contribution:** 2 fair
**Rating:** 5
**Confidence:** 4

**Summary:**

The paper considered a multiple transport problem with applications in visual matching, model fusion. More particular, instead of working on OT problem between distributions, the paper considered  a sequence of ordered distributions and find the transport plan such that its circular compositions of transportation map is actually the identity map.  The authors added another regularized  function to deal with the cycle-consistency requirement, then use the entropic approach of Cuturi to deploy Sinkhorn algorithm to estimate its solution. The authors applied the methods to solve multi-point matching and multi-model fusion and demonstrated their methods for several datasets, i.e Pascal VOC, Willow, MNIST and CIFAR-10.

**Strengths:**

It appears to be interesting theoretical variation of OT problem.
The authors  derived an algorithm to find its solution through the connection with the Sinkhorn algorithm.
The experiment results show a slightly improvement to the one of using Sinkhorn algorithm.

**Weaknesses:**

Performance of the proposed method is incremental improvement to that of using Sinkhorn without the cycle consistency requirement.

In comparison with Sinkhorn's method, RMOT-Sinkhorn just have one more constraint for the last mapping. In fact, it is not much different from the one using Sinkhorn.

I do not agree with other two metrics, i.e CR and CACC,  assessing the performance of methods, since they favor the author's method.

In the problem of multi-matching and model fusion, I do not see that the method is natural application to those problems unless the data sets have cyclical structures. From Figure 5 for example, I do not see the data have that cyclical structure, if we switch the order  of pictures of the second column and third column, does it change the final results? The most natural applications, among those presented, of the method is TSP, but the empirical result is still worse than its current SOTA result.

With the cycle consistency requirement, the problem wants a solution which is a permutation matrix at every transportation stage, since product of "transition" matrices, (all entries are non-negative),  must be equal to identity matrix. Hence the real constraint is to make the transport maps are close to permutation.  The objective function (17) is just a relaxed version of the true objective function. What is the guideline for choosing the parameter $\delta$ and $\epsilon$?

Overall, it appears to be interesting problem. But it also appears to be a slightly different version of multi-OT problems, thus it looks like an incremental solution to some current methods. The contribution to OT theory is also limited, i.e no proof of convergence property etc.

**Questions:**

In table 1, why does the Sinkhorn perform better than RMOT-Sinkhorn in PCA-GM in the AC index?

In table 1, column NMGM, we have CR $100\%$ but the ACC is only equal to $93.76\%$. Does it mean that the consistent rate is not a good indicator of  accuracy  rate?

---

> ### Author Response · Authors · 2023-11-19
>
> #### Q1:  In comparison with Sinkhorn's method, RMOT-Sinkhorn just has one more constraint for the last mapping. In fact, it is not much different from the one using Sinkhorn.
>
> #### A1: We address the question with following three points: 1) Many experimental results demonstrate that our RMOT-Sinkhorn algorithm significantly improves upon the pairwise Sinkhorn algorithm which is evident in the enhanced performance of CR and CACC metrics and most cases in ACC evaluation. 2) There may exist some misunderstandings that RMOT-Sinkhorn is not only for the last mapping. All the mapping will be affected (e.g. results in Fig.3).   3) RMOT-Sinkhorn is an iterative Sinkhorn method, which is similar to calculating the entropic Gromov-Wasserstein Distance. The key lies in effectively and reasonably calculating the modification of the cost matrix, which is not a straightforward task.
>
>
> #### Q2: I do not agree with other two metrics, i.e CR and CACC, assessing the performance of methods, since they favor the author's method.
>
> #### A2: In fact, similar metrics to CR have been widely used in literature in terms of multiple objects matching, e.g. specifically in Fig. 4(e) of [1] and the reference therein e.g. [2] which could be regarded as a special case of ours when everytime the number of sets for consistency computing is three. Moreover, CR and CACC can be regarded as extensions and refinements of ACC for multiple sets. They have meaningful and interpretable implications. We can redefine ACC from a matrix computation perspective as follows:
> $$
>    ACC = \frac{1}{n} <\prod_{k=1}^2(\hat{\mathbf{P}}_k\odot \mathbf{Y}_k),\mathbf{I}_n>
> $$
> where $\mathbf{Y}_k$ represents the ground truth as a one-hot matrix. $\hat{\mathbf{P}}_1$ is the probability matrix for matching from the first set of points to the second set, and $\hat{\mathbf{P}}_2$ is the probability matrix for matching from the second set to the first set. In this particular case, we have $\hat{\mathbf{P}}_1= \hat{\mathbf{P}}_2^\top$ and $\mathbf{Y}_1=\mathbf{Y}_2^\top$. From this perspective, we define the CR and CACC metrics, incorporating the concept of K-marginal transportation:
> $$
>     \text{CR} = \frac{1}{n}<\prod_{k=1}^K\hat{\mathbf{P}}_k,\mathbf{I}_n> \text{ and } \text{CACC}=\frac{1}{n}<\prod_{k=1}^K(\hat{\mathbf{P}}_k\odot \mathbf{Y}_k),\mathbf{I}_n>
> $$
> Note that $CR$ refers to the accuracy of forming cycles through matching, capturing the ability to establish correct pairwise matches within the set of $K$ points. On the other hand, CACC represents the accuracy of correctly matching all $K$ points, taking into account the entire cycle instead of only considering pairs of points as in traditional ACC metrics.
>
>
> #### Q3: I do not see that the method is natural application to those problems unless the data sets have cyclical structures (e.g. Switching the order of Fig. 5).
>
> #### A3: #### A3: Our model can be applied to various datasets [3], particularly those involving multi-modal data for matching. The reviewer may be concerned about the choice of order because natural datasets often exhibit more of a pairwise structure rather than a directed cyclical structure. However, this does not affect the usability of our model. To illustrate, consider three sets: $A$, $B$, and $C$, with a mapping order defined as $A\to B$, $B\to C$, and $C\to A$, represented by matching matrices $\mathbf{P}_1$, $\mathbf{P}_2$, and $\mathbf{P}_3$. Consistency requires that $\mathbf{P}_1\mathbf{P}_2\mathbf{P}_3=\mathbf{I}$. If we switch the order to $A\to C$, $C\to B$, and $B\to A$, with matching matrices $\mathbf{P}_3^\top$, $\mathbf{P}_2^\top$, and $\mathbf{P}_1^\top$, the consistency becomes $\mathbf{P}_3^\top\mathbf{P}_2^\top\mathbf{P}_1^\top=\mathbf{P}_1\mathbf{P}_2\mathbf{P}_3=\mathbf{I}$. Therefore, for the case of three measures ($K=3$), switching the order (of Fig. 5) does not affect the problem formulation.
>
> For $K>3$, switching the order does indeed impact the formulation of the problem for datasets that exhibit pairwise relationships. However, note our directed cyclical structure is essentially a subgraph of pairwise structure. When the latter is satisfied, the former is also satisfied, allowing our method to still improve the matching performance. Besides, we can select multiple directed cyclical structures for OMT to achieve equivalence of pairwise structure one.

---

> ### Author Response · Authors · 2023-11-19
>
> #### Q4: the guideline for choosing the parameter $\delta$ and $\epsilon$.
>
> #### A4: We search for the optimal values of $\delta$ and $\epsilon$ in the validation dataset by maximizing the mean values of ACC and CACC metrics. Once determined, we apply these values to the testing data for evaluation.
>
> #### Q5: it also appears to be a slightly different version of multi-OT problems, thus it looks like an incremental solution to some current methods.
>
> #### A5: The key of Multi-marginal OT is the definition of cost tensor and if one can define the cost function among multiple marginals with consistency constraints, our OMT will become a subproblem of Multi-marginal OT. However, for the calculation, Multi-Marginal OT has an exponential increase in spatial complexity (N^K) as the number of marginals increases, while our method has a linear increase (NK), which is also a main difference.
>
> #### Q6: In table 1, why does the Sinkhorn perform better than RMOT-Sinkhorn in PCA-GM in the ACC index?
>
> #### A6: The regularization introduced by RMOT-Sinkhorn essentially increases the probability of forming cycles, ensuring that any given feature starts from the origin and returns to the origin. However, it does not guarantee an increase in the accuracy of each individual match within the cycle. For example, let's consider three sets: $A$, $B$, and $C$. Starting with the first feature $A[1]$, the prediction of Pairwise Sinkhorn is $A[1]\to B[2]$, $B[2]\to C[2]$, and $C[2]\to A[2]$. The modification introduced by RMOT-Sinkhorn could result in a change of matches to $A[2]\to B[2]$, $B[2]\to C[2]$, and $C[2]\to A[2]$. However, it is also possible for the modification to change the matches to $A[1]\to B[2]$, $B[2]\to C[2]$, and $C[2]\to A[1]$. The latter case actually decreases the evaluation of ACC.
>
> #### Q7: In table 1, column NMGM, we have CR=100 but the ACC is only equal to 93.76. Does it mean that the consistent rate is not a good indicator of accuracy rate?
>
> #### A7: $CR=100\%$  indicates that all the matches form cycles, but it does not guarantee that every individual match within the cycle is correct. On the other hand, a $CACC$ of $86.82%$ means that there are errors in one or more nodes within $13.18%$ of the cycles. Furthermore, one advantage of evaluating with $CR$ is that it does not require supervised data and it considers whether the predictions themselves are logically consistent, which is a strength of this evaluation metric.
>
> BTW, it has been widely recognized that consistency has an empirically strong correlation to accuracy in many public benchmarks e.g. [1,2]. We will also add more references to support this point.
>
> [1] Unifying Offline and Online Multi-Graph Matching via Finding Shortest Paths on Supergraph, IEEE TPAMI 2021
> [2] Multi-Graph Matching via Affinity Optimization with Graduated Consistency Regularization, IEEE TPAMI 2016
> [3] Artificial intelligence for multimodal data integration in oncology. Cancer cell 40, no. 10 (2022): 1095-1110.

---

> > ### Comment · Reviewer_16yx · 2023-11-23
> > **Response to the rebuttal**
> >
> > Dear authors,
> >
> > I would like to thank authors for their detailed and lengthy answers. Similar to the reviewer aQyc, I think the criteria CACC and CR are biased to the presented method. For the CACC, its function is similar to the penalty function, namely, including product terms of "transition matrices". For the CR, it obtains 100% but the corresponding accuracy is only around 90%. Thus, I only trust ACC to compare the effectiveness between methods. Moreover, the ACC of ROMT-Sinkhorn are not always better than that of Sinkhorn alone (table 1, PCA-GM), even ROMT-Sinkhorn has extra parameter $\delta$ to tune.
> >
> > Hence, I would like to keep my score unchanged. I  believe there are a lot of room to improve the results theoretically and empirically. I agree with other reviewers that this version is not ready for publication.

---

> ### Author Response · Authors · 2023-11-22
>
> Dear Reviewer,
>
> Thank you for taking the time to review our paper. We greatly appreciate your valuable feedback and insightful comments. As the discussion deadline approaches, we would like to request your feedback on our response to address the concerns. We are also open to any further questions or suggestions that you may have.
>
> Thanks for your time and attention. Your expertise and guidance are invaluable for our research.
>
> Best regards,
>
> The authors

---

### Official Review · Reviewer_2Gzs · 2023-11-01

**Soundness:** 2 fair
**Presentation:** 3 good
**Contribution:** 2 fair
**Rating:** 5
**Confidence:** 4

**Summary:**

This paper introduced the Monge and Kantorovich formulations of a novel Optimal Transport problem, termed Optimal Multiple Transportation (OMT), between more than two probability measures. OMT minimizes the total pairwise transportation costs while ensuring cycle consistency. By adding the entropic and cycle consistency regularization, OMT can be efficiently solved by an iterative Sinkhorn algorithm, termed ROMT-Sinkhorn. As a side product, the authors developed a new formulation for the Traveling Salesman Problem (TSP). The newly introduced problem, TSP-OMT, can be solved greedily in the probability space instead of the conventional distance space. Finally, empirical results showed the effectiveness of the proposed ROMT-Sinkhorn algorithm in two applications: multi-point matching and multi-model fusion.

**Strengths:**

- To the best of my knowledge, the introduced OT problem in this paper is novel.
- This paper is well-written and easy to follow.
- The authors clearly distinguished between Multi-marginal OT and OMT.
- The cycle consistency requirement is well-motivated in applications.
- The effectiveness of the proposed RMOT-Sinkhorn is supported by the empirical results to some extent.

**Weaknesses:**

- The literature review for model fusion is not well-written.
- This paper lacks some theoretical results.
- My main concern is the scalability of OMT-Sinkhorn. All examples/experiments only consider three measures.

**Questions:**

- The literature for Model Fusion lacks a lot of related papers. Here are a few recent papers on the topic of model fusion:
  - [r1] Hongyi Wang, Mikhail Yurochkin, Yuekai Sun, Dimitris Papailiopoulos, and Yasaman Khazaeni. Federated learning with matched averaging. In International Conference on Learning Representations, 2020.
  - [r2] Mitchell Wortsman, Gabriel Ilharco, Samir Ya Gadre, Rebecca Roelofs, Raphael Gontijo-Lopes, Ari S Morcos, Hongseok Namkoong, Ali Farhadi, Yair Carmon, Simon Kornblith, et al. Model soups: averaging weights of multiple fine-tuned models improves accuracy without increasing inference time. In International Conference on Machine Learning, pp. 23965–23998. PMLR, 2022.
  - [r3] Michael S Matena and Colin A Raffel. Merging models with fisher-weighted averaging. Advances in Neural Information Processing Systems, 35:17703–17716, 2022.
  - [r4] Akash, Aditya Kumar, Sixu Li, and Nicolás García Trillos. "Wasserstein Barycenter-based Model Fusion and Linear Mode Connectivity of Neural Networks." arXiv preprint arXiv:2210.06671 (2022).
  - [r5] Ainsworth, Samuel K., Jonathan Hayase, and Siddhartha Srinivasa. Git re-basin: Merging models modulo permutation symmetries. In International Conference on Learning Representations, 2023.
  - [r6] Dang Nguyen, Trang Nguyen, Khai Nguyen, Dinh Phung, Hung Bui, and Nhat Ho. On cross-layer alignment for model fusion of heterogeneous neural networks. In ICASSP 2023-2023 IEEE International Conference on Acoustics, Speech and Signal Processing (ICASSP), pp. 1–5. IEEE, 2023.
  - [r7] Stoica, George, Daniel Bolya, Jakob Bjorner, Taylor Hearn, and Judy Hoffman. "ZipIt! Merging Models from Different Tasks without Training." arXiv preprint arXiv:2305.03053 (2023).
  - [r8] Imfeld, Moritz, Jacopo Graldi, Marco Giordano, Thomas Hofmann, Sotiris Anagnostidis, and Sidak Pal Singh. "Transformer Fusion with Optimal Transport." arXiv preprint arXiv:2310.05719 (2023).
- Can the authors provide the complexity analysis of OMT-Sinkhorn?
- Eq. 15: “We set $\delta_k > 0$ for $k < K$ to make  $(P_k)\_{ii}$ approach 0 for every $k < K$, and $\delta_K < 0$ to make $(P_k)_{ii}$ approach 1.” Should the coefficients be set reversely?
- **Scalability**. Can the authors demonstrate the applications for more measures? Like in OTFusion (Singh & Jaggi, 2020), the authors can fuse 4 and 6 models. In addition, the scale of the architecture in model fusion is also relatively small. Model fusion can fuse ResNet, LSTM, and even Transformer.
- The number of baselines is quite limited. One possible baseline can be generated from the optimal solution of Multi-marginal OT. Given $P$ is the optimal solution for multi-marginal OT. We can have $P_1 = \sum_{i_3 = 1}^{n_3} P_{i_1, i_2, i_3}$. $P_2$ and $P_3$ can be defined similarly. Another simple baseline is to iteratively fuse two models.
- Tab. 3: For CIFAR10 + VGG11, the finetuned accuracy is the same as the best pre-trained one, which raises concerns about efficiency.

**Minor**:
- The title of Section 2.3 should be only “Visual Point Matching and Model Fusion” because the usage of OT is not mentioned enough.
- Notation consistency for $\mathbb{R}^{+}$.
- Should use different notations for $\mathcal{C}$ in Eq. (7) and (8).
- entropy regularization → entropic regularization
- The usage of $\delta$ as a regularization coefficient should be avoided as $\delta$ denotes the Dirac measure earlier.
- “Tab. 3 shows the fusion results” This sentence can be removed from Section 3.4.
- Some typos

---

> ### Author Response · Authors · 2023-11-19
>
> ##### Q1. The literature review for model fusion is not well-written.
>
> ##### A1: Thanks for your information and we will add these references. The revised literature review is uploaded in the new version.
>
> ##### Q2:  My main concern is the scalability of OMT-Sinkhorn. All examples/experiments only consider three measures.
>
> ##### A2:  We add visual matching experiments of four and five measures for OMT and the results(\%) are shown as follows:
>
> | Willow_4GM                   | ACC        | CACC       | CR         |
> | ---------------------------- | ---------- | ---------- | ---------- |
> | IPCA-GM-Hungarian            | 0.3924     | 0.1610     | 0.5473     |
> | IPCA-GM-EMD                  | 0.9383     | 0.8221     | 0.8775     |
> | IPCA-GM-Sinkhorn             | 0.9381     | 0.8214     | 0.8693     |
> | IPCA-GM-ROMT-Sinkhorn (ours) | **0.9453** | **0.8592** | **0.9240** |
>
>
> | Willow_5GM                   | ACC        | CACC       | CR         |
> | ---------------------------- | ---------- | ---------- | ---------- |
> | IPCA-GM-Hungarian            | 0.3973     | 0.1331     | 0.4888     |
> | IPCA-GM-EMD                  | 0.9401     | 0.7927     | 0.8612     |
> | IPCA-GM-Sinkhorn             | 0.9402     | 0.7881     | 0.8493     |
> | IPCA-GM-ROMT-Sinkhorn (ours) | **0.9460** | **0.8322** | **0.9142** |
>
> The experiment utilized IPCA-GM with VGG16 as the backbone and evaluated its performance on the WillowObject dataset. Besides, for **four** model fusion, the results(\%) of MNIST+MLP (with fine-tuning) are given as follows :
>
> | model A| model B | model C | model D | PREDICTION | VANILLA | OTfusion | Ours |
> | ------- | ------- | ------- | ------- | ---------- | ------- | -------- | ---- |
> |  97.72  | 97.75   | 97.69   | 97.26   | 97.91      | 97.86   | 98.08    | 98.08 |
> The fusion for different models e.g. ResNet, LSTM, and even Transformer are not finished.  We'll get it done as soon as possible.
>
>
> #### Q2: Baseline for Multi-Marginal OT.
>
> #### A2: We include Multi-Marginal OT as baselines in the visual matching experiments, where the cost tensor is defined as $$C_{ijk} = C_{ij}+C'_{jk}+C''_{ki}$$, with cost matrices $C, C', C''$. The Bregman iterative method [1] is employed for calculation. Following your advice, we utilize $\sum_k P_{ijk}$ as the probability for the transportation between $i$ and $j$, and similarly for $\sum_j P_{ijk}$ and $\sum_i P_{ijk}$. The experimental results are presented as follows:
>
> | Willow_3GM                   | ACC        | CACC       | CR         |
> | ---------------------------- | ---------- | ---------- | ---------- |
> | IPCA-GM-Hungarian            | 0.4003     | 0.2269     | 0.6303     |
> | IPCA-GM-EMD                  | 0.9426     | 0.8732     | 0.9141     |
> | IPCA-GM-Sinkhorn             | 0.9451     | 0.8776     | 0.9144     |
> | IPCA-GM-MMOT                 | 0.9422     | 0.8793     | 0.9057     |
> | IPCA-GM-ROMT-Sinkhorn (ours) | **0.9506** | **0.9031** | **0.9500** |
>
> | Willow_4GM                   | ACC        | CACC       | CR         |
> | ---------------------------- | ---------- | ---------- | ---------- |
> | IPCA-GM-Hungarian            | 0.3924     | 0.1610     | 0.5473     |
> | IPCA-GM-EMD                  | 0.9383     | 0.8221     | 0.8775     |
> | IPCA-GM-Sinkhorn             | 0.9381     | 0.8214     | 0.8693     |
> | IPCA-GM-MMOT                 | 0.9398     | 0.8268     | 0.8575     |
> | IPCA-GM-ROMT-Sinkhorn (ours) | **0.9453** | **0.8592** | **0.9240** |
>
> The experimental setup for this experiment is the same as that described in Section A2. In terms of method comparison, Multi-Marginal OT has an exponential increase in spatial complexity ($O(N^2K)$) as the number of marginals increases, while our method has a linear increase ($O(N^2K)$). Note $N$ is the marginal dimension number (assuming the same) and $K$ is the number of marginal.
>
> #### Q3: In Tab. 3, for CIFAR10 + VGG11, the finetuned accuracy is the same as the best pre-trained one, which raises concerns about efficiency.
>
> #### A3: In Table 3, our OMT fusion method is compared to the OTfusion method, and it achieves better performance. However, it should be noted that not all fusion methods result in improved performance compared to the original models. It is normal for the fused model to have lower accuracy than the original model, especially in the case of multi-model fusion. The same accuracy between our method and the original one is actually a coincidence, as their actual prediction results are different.
>
> #### Q4: Minors and typos
>
> #### A4: Thank you for the reminder. We have made the corrections in the new PDF version. Thanks again.
>
> [1] Benamou, Jean-David, Guillaume Carlier, Marco Cuturi, Luca Nenna, and Gabriel Peyré. "Iterative Bregman projections for regularized transportation problems." SIAM Journal on Scientific Computing 37, no. 2 (2015): A1111-A1138.

---

> ### Author Response · Authors · 2023-11-22
>
> Dear Reviewer,
>
>   Thank you for your valuable response. As the discussion deadline approaches, we would like to request your feedback on our response to address the concerns. We are also open to any further questions or suggestions that you may have.
>
> Thanks for your time and attention. Your expertise and guidance are invaluable for our research.
>
> Best regards,
>
> The authors

---

> > ### Comment · Reviewer_2Gzs · 2023-11-22
> > **Response to authors' rebuttal**
> >
> > I thank the authors for reading and responding to all my questions. However, I found that the answer to the time complexity question is not well-addressed. In addition, some experimental settings (e.g., baseline descriptions and hyperparameter settings) are missing.
> > - Baselines: "We compare our ROMT-Sinkhorn algorithm with (Munkres, 1957), EMD (Dantzig, 1949) and Sinkhorn Algorithm (Cuturi, 2013)." -> Hungarian and MMOT are missing. While I understand what they are, the authors need to write them explicitly.
> > - Hyperparameters: What are the choices of L, $\epsilon$, $\delta$ in the experiments?
> > - The training procedures are also missing.
> >
> > Further note: I hope that the authors take their time to write neat answers next time.
> >
> > I would like to keep my original score and agree with *Reviewer aQyc* that this manuscript is not ready for publication.

---

### Author Response · Authors · 2023-11-19
**General Response**

Dear Area Chairs and Reviewers,

We sincerely appreciate the reviewers for investing their valuable time and providing insightful comments on our paper. Overall, the reviewers found our work to be novel (2Gzs, RCtv, aQyc), interesting (16yx, uVEA, aQyc), well-written and easy to follow (2Gzs, uVEA). Additionally, they find our OMT is clear and well-motivated (2Gzs), generalized and significant (uVEA),  and it is interesting to see the connection between OMT and TSP (uVEA, aQyc).

In the author response period, we make every effort to address reviewers' concerns and provide additional experimental results to further verify our contribution. The summary of our efforts is presented as follows:

1) We conduct more experiments for considering more than three measures and the results are given in Tab. 2, Tab. 5 (for visual matching), and Tab.6 (for model fusion).

2) We add Multi-marginal OT as baseline and the results are given in Tab.1, Tab.2, Tab.3, and Tab.5.

3) We update the related work for model fusion in Subsection 2.3.

4) We further explain the issue of order switching. When K=3 (i.e., the case of three measures), the problem is theoretically equivalent. When K>3, exchanging the measure order has minimal impact on the prediction results. The experimental results are provided in Table 8 and Tab. 9.

5) We explain the selection of hyperparameters $\epsilon$ and $\delta$. Based on the validation set, we choose the best-performing $\epsilon$ and $\delta$ values and use them in the testing data for matching inference. Additionally, we conducted an Ablation Study for $\epsilon$ and $\delta, as shown in Tab. 10.

6) We discuss the complexity of OMT, and a comparison of the running time is provided in Tab. 10.

7) Regarding the evaluation metrics CACC and CR, we explain that similar metrics have also been applied in previous works. These two metrics can be seen as extensions of ACC for multi-point matching and have practical physical significance.

Once again, we would like to express our gratitude to the Area Chairs and reviewers for their time and effort invested in reviewing our work.

---

### Author Response · Authors · 2023-11-20
**Response to Reviewers**

Dear reviewers,

We would like to express our sincere gratitude again for your valuable comments and thoughtful suggestions. Throughout the rebuttal phase, we tried our best to address concerns, augment experiments to fortify the paper (comprising approximately 3 pages of new content with 6 new tables) and refine details in alignment with your constructive feedback. Since the discussion time window is very tight and is approaching its end, we truly hope that our responses have met your expectations and assuaged any concerns. We genuinely do not want to miss the opportunity to engage in further discussions with you, which we hope could contribute to a more comprehensive evaluation of our work. Should any lingering questions persist, we are more than willing to offer any necessary clarifications.

With heartfelt gratitude and warmest regards,

The Authors

---

### Author Response · Authors · 2023-11-22

Dear AC,

  as all the ratings are at the borderline, could you help launch the discussion among the reviewers, and we are looking forward to further comments and questions. We are more than happy to respond.

The authors.

---

### Meta-Review · Area_Chair_j7yp · 2023-12-06

**Metareview:**

All reviewers acknowledged that the paper presents an interesting algorithmic contribution, notably leveraging a novel connection to the TSP. Additionally, the extensive numerical evaluations conducted were appreciated. The authors have also provided a strong rebuttal, enriched by the inclusion of new numerical results. However, it is unanimously agreed upon by all reviewers that the paper is not yet ready for publication in its current form. A significant concern is that the theoretical results appear somewhat limited. Reviewers have raised important questions regarding the convergence of the proposed methods and their connection with existing proximal algorithms, indicating areas that require deeper exploration and clarification. Furthermore, it is suggested that the inclusion of additional numerical results, particularly those involving common baselines, would be crucial in providing a more comprehensive evaluation of the proposed approach.

**Justification For Why Not Higher Score:**

Numerics + theory would be needed to pass the threshold.

**Justification For Why Not Lower Score:**

N/A

---

### Decision · Program_Chairs · 2024-01-16

Reject